# Low-rank Matrix Bandits with Heavy-tailed Rewards

Yue Kang[1]        Cho-Jui Hsieh[2]        Thomas C. M. Lee[1]

[1]Department of Statistics, UC Davis, Davis, CA, USA
[2]Google and Department of Computer Science, UCLA, Los Angeles, CA, USA

## Abstract

In stochastic low-rank matrix bandit, the expected reward of an arm is equal to the inner product between its feature matrix and some unknown $d_1$ by $d_2$ low-rank parameter matrix $\Theta^*$ with rank $r \ll d_1 \wedge d_2$. While all prior studies assume the payoffs are mixed with sub-Gaussian noises, in this work we loosen this strict assumption and consider the new problem of low-rank matrix bandit with heavy-tailed rewards (LowHTR), where the rewards only have finite $(1 + \delta)$ moment for some $\delta \in (0, 1]$. By utilizing the truncation on observed payoffs and the dynamic exploration, we propose a novel algorithm called LOTUS attaining the regret bound of order $\tilde{O}(d^{\frac{3}{2}} r^{\frac{1}{2}} T^{\frac{1}{1+\delta}} / \tilde{D}_{rr})$[1] without knowing $T$, which matches the state-of-the-art regret bound under sub-Gaussian noises [Lu et al., 2021, Kang et al., 2022] with $\delta = 1$. Moreover, we establish a lower bound of the order $\Omega(d^{\frac{\delta}{1+\delta}} r^{\frac{\delta}{1+\delta}} T^{\frac{1}{1+\delta}}) = \Omega(T^{\frac{1}{1+\delta}})$ for LowHTR, which indicates our LOTUS is nearly optimal in the order of $T$. In addition, we improve LOTUS so that it does not require knowledge of the rank $r$ with $\tilde{O}(dr^{\frac{3}{2}} T^{\frac{1+\delta}{1+2\delta}})$ regret bound, and it is efficient under the high-dimensional scenario. We also conduct simulations to demonstrate the practical superiority of our algorithm.

---

[1]$\tilde{O}$ ignores polylogarithmic factors. We denote $d := d_1 \vee d_2$ and $\tilde{D}_{rr} := (D_{rr} - 1)\mathbb{1}_{\delta=1} + 1$ where $D_{rr}$ is the $r$-th singular value of $\Theta^*$.

## 1  INTRODUCTION

The Multi-armed Bandit (MAB) has proven to be a powerful framework to model various decision-making problems with great applications to medical trials [Villar et al.,

2015], personalized recommendation [Li et al., 2010], and hyperparameter learning [Ding et al., 2022, Kang et al., 2024], etc. To leverage the side information (contexts) of arms in real-world scenarios, the most important variant of MAB, named stochastic linear bandit (SLB), has been extensively investigated. However, the rise of high-dimensional sparse data in modern applications [Zou, 2006, Han and Lee, 2022] has revealed the inefficiencies of the traditional SLB, particularly in its failure to account for sparsity. To address this limitation, the stochastic high-dimensional bandit with low-dimensional structures has emerged as the pioneering model, such as the LASSO bandit [Bastani and Bayati, 2020] and the low-rank matrix bandit [Jun et al., 2019]. In this work, we investigate the stochastic low-rank matrix bandit, where at each round $t$ the agent first observes the arm set $\mathcal{X}_t \subseteq \mathbb{R}^{d_1 \times d_2}$ composing of context matrices ($\mathcal{X}_t$ can be infinite and changing over time). Then the agent pulls an arm $X_t \in \mathcal{X}_t$ and only obtains its associated noisy reward $y_t = \langle X_t, \Theta^* \rangle + \eta_t$ with some inherent low-rank parameter $\Theta^*$ and zero-mean white noise $\eta_t$. This bandit problem is broadly applicable in recommendation systems with pair contexts, like dating service and combined flight-hotel promotion [Kang et al., 2022].

In all existing literature on low-rank matrix bandit, a default assumption is that the noise $\eta_t$ is sub-Gaussian conditioned on historical observations [Jun et al., 2019]. However, in various real-world scenarios such as financial markets [Bradley and Taqqu, 2003, Cont and Bouchaud, 2000], there's a notable trend where extreme noise, a.k.a. heavy-tailed noise, in observations occur more frequently than what would be expected under a sub-Gaussian distribution, in which case previous studies would become futile. These heavy-tailed observations do not exhibit exponential decay and may crucially affect the estimation. To address this challenge, a line of algorithms has been proposed to handle heavy-tailed noise under MAB [Bubeck et al., 2013] and SLB [Medina and Yang, 2016]. However, to the best of our knowledge, effectively managing heavy-tailed noise under the more complex and efficient low-rank matrix bandit framework

remains unexplored. In this study, we examine this crucial problem: low-rank matrix bandit with heavy-tailed rewards (LowHTR). Specifically, to keep consistent with the heavy-tailed studies under MAB and SLB, we assume that the noise has finite $(1 + \delta)$ moment for some $\delta \in (0, 1]$. We first propose an efficient algorithm named LOTUS when $T$ is unrevealed to the agent. Then we demonstrate it attains a regret lower bound of LowHTR for the order of $T$ ignoring logarithmic factors. Our LOTUS can be further improved to be agnostic to rank $r$ with slightly worse regret bound.

The detailed contributions of our work can be summarized as follows: (1) inspired by the success of Huber loss [Kang and Kim, 2023, Sun et al., 2020] and nuclear norm penalization [Negahban and Wainwright, 2011], we first revisit the convex-relaxation-based estimator to approximate the low-rank parameter matrix with heavy-tailed noise. As far as we're aware, our work is the first one to solve the trace regression problem under arbitrary heavy-tailed noise with bounded $(1 + \delta)$ moment ($\delta \in (0, 1)$), which is highly non-trivial and stands as a noteworthy advancement on its own merits. (2) Equipped with the aforementioned estimator, we develop an algorithm named LOTUS for LowHTR. LOTUS exploits the estimated subspace by proposing a sub-method called LowTO that extends from the TOFU algorithm [Shao et al., 2018] designed for SLB with heavy-tailed noise. Our LowTO truncates the rewards to mitigate the heavy-tailed effect and penalizes the redundant features within the sparsity structure. When the total horizon $T$ is unrevealed, our algorithm could adaptively switch between exploration and exploitation to achieve the $\tilde{O}(d^{\frac{3}{2}} r^{\frac{1}{2}} T^{\frac{1}{1+\delta}} / \tilde{D}_{rr})$ regret bound. (3) We further provide a lower bound for LowHTR of order $\Omega(d^{\frac{\delta}{1+\delta}} r^{\frac{\delta}{1+\delta}} T^{\frac{1}{1+\delta}})$, which indicates that our LOTUS is nearly optimal in the scale of $T$. (4) While all existing works on low-rank matrix bandits require a priori knowledge of the rank $r$, we further improve our LOTUS to operate without knowing $r$ even under the more difficult heavy-tailed setting with $\tilde{O}(dr^{\frac{3}{2}} T^{\frac{1+\delta}{1+2\delta}} + d^{\frac{3}{2}} r^{\frac{1}{2}} T^{\frac{1}{1+\delta}})$ regret bound, which is better than the trivial one in high-dimensional case, i.e. when $d \gtrsim T^{\frac{\delta^2}{(1+2\delta)(1+\delta)}}$. Intuitively, it obtains a useful rank $\hat{r}$ by truncating the estimated singular values at each batch. (4) The practical superiority of our LOTUS is then firmly validated in our simulations.

**Notations:** For any vector $x \in \mathbb{R}^n$, we use $\|x\|_p$ to denote the $l_p$-norm of the vector $x$ and $\|x\|_H = \sqrt{x^\top H x}$ to denote its weighted 2-norm with regard to some positive definite matrix $H \in \mathbb{R}^{n \times n}$. For matrices $X, Y \in \mathbb{R}^{n_1 \times n_2}$, we use $\|X\|_{\text{op}}$, $\|X\|_{\text{nuc}}$ and $\|X\|_{\text{F}}$ to define the operator norm, nuclear norm and Frobenious norm of the matrix $X$ respectively, and we write $\langle X, Y \rangle := \text{trace}(X^\top Y)$ as their inner product. We also write $f(n) \asymp g(n)$ if $f(n) = O(g(n))$ and $g(n) = O(f(n))$, $f(n) \gtrsim g(n)$ if $g(n) = O(f(n))$, and $f(n) \lesssim g(n)$ if $f(n) = O(g(n))$, and these are the common notations used in the high-dimensional statistics

literature [Wainwright, 2019].

## 2 RELATED WORK

**Bandit under Heavy-tailedness** Research on bandits with heavy-tailed rewards assumes the noise has finite $(1 + \delta)$ moment, $\delta \in (0, 1)$, and most existing algorithms follow two key strategies: truncation and median of means. Start with Bubeck et al. [2013], a UCB-based algorithm was proposed for MAB with heavy-tailed rewards, enjoying a logarithmic regret bound. To extend their study to the SLB setting, Medina and Yang [2016] developed two algorithms based on the truncation and median of means ideas, but both methods could only attain the regret bound of order $\tilde{O}(T^{\frac{3}{4}})$ when $\epsilon = 1$, which fails to fulfill our expectations. Shao et al. [2018] then refined their results on SLB and introduced two algorithms with improved regret bound. They also constructed a matching lower bound with $T$. Xue et al. [2020] investigated on the finite arm case and provided two SubLinUCB-based [Chu et al., 2011] algorithms. Recently, Kang and Kim [2023] borrowed the ideas from Huber regression and proposed an improved Huber bandit under finite arm sets. However, their work is confined to the low-dimensional bandit without sparsity, and their parameter vectors are presumed to be arm-dependent under the finite arm set. Another contemporary work Xue et al. [2023] developed a nearly optimal algorithm for arbitrary arm sets with reduced computation in practice. Yet, none of these studies tackle the heavy-tailedness under the more challenging contextual high-dimensional bandits problem with sparsity, a useful niche our work aims to fill.

**Low-rank Matrix Bandit** There has been a line of literature on stochastic low-rank matrix bandit with sub-Gaussian noise. Initially, Jun et al. [2019] introduced the bilinear low-rank matrix bandit problem and proposed the two-stage ESTR algorithm with $\tilde{O}(\sqrt{d^3 rT}/D_{rr})$ regret bound. Jang et al. [2021] then constructed a new algorithm improving the regret bound by $\sqrt{r}$. Lu et al. [2021], Kang et al. [2022] extended the problem setting to low-rank matrix bandit where feature matrices no longer have to be rank-one. Specifically, Lu et al. [2021] first proposed the LowGLOC with $\tilde{O}(\sqrt{d^3 rT})$ regret bound, but this method is computationally prohibitive and cannot handle the contextual setting. Subsequently, Lu et al. [2021], Kang et al. [2022] developed several more efficient algorithms, achieving regret bound of order $\tilde{O}(\sqrt{d^3 rT}/D_{rr})$. Our work broadens this research scope to encompass arbitrary heavy-tailed noise with bounded $(1+\delta)$ moment ($\delta \in (0, 1)$), and our algorithm LOTUS obtains the $\tilde{O}(d^{\frac{3}{2}} r^{\frac{1}{2}} T^{\frac{1}{1+\delta}} / \tilde{D}_{rr})$ regret bound, which coincides with the aforementioned leading one with $\delta = 1$. Moreover, we showcase that our regret bound is optimal concerning the order of $T$ with a matching lower bound. Another notable limitation in existing algorithms for low-rank matrix bandits is their dependence on the rank $r$, which

is impractical. We further improve our LOTUS method to be agnostic to $r$ with a slightly worse regret bound, which represents the first attempt at this real-world issue. Jang et al. [2024] recently proposed a new estimator utilizing the geometry of the arm set to conduct estimation.

**Matrix Recovery under Heavy-tailedness**   All studies on low-rank matrix estimation revolve around two ideas: Convex approaches tend to replace the classic square loss with some more robust ones, like the renowned Huber loss [Huber, 1965, Sun et al., 2020]. Tan et al. [2022] considered the sparse multitask regression under heavy-tailed noise, contrasting our focus on the trace regression problem. The two works most closely related to ours are Fan et al. [2021], Yu et al. [2023]. Fan et al. [2021] established a two-step method for the robust trace regression, but they assumed the noise possesses finite $2k$ moment for $k > 1$ and their approximation error is not even proportional to the noise size. Yu et al. [2023] further employed the Huber loss to develop an enhanced regressor with error aligned with the noise scale as long as the noise has bounded variance. In our work, we further complement their result and utilize the Huber-type estimator robust to noise with only finite $(1 + \delta)$ moment for any $\delta \in (0, 1]$, and we deduce the error rate of order $\tilde{O}((d/n)^{\frac{\delta}{1+\delta}} \mathbb{E}(|\eta_t|^{1+\delta})^{\frac{1}{1+\delta}})$ scaling with the noise scale decently. On the other hand, nonconvex methods aim to seek local optima of the matrix recovery problem via gradient descent. The notable work [Shen et al., 2022] developed a Riemannian sub-gradient method and attained the optimal statistical rate under heavy-tailed noises with bounded $(1 + \delta)$ moment, but their work relies on some additional assumptions like the noise is symmetric or zero-median. In summary, our work stands as the first solution to address the trace regression problem under arbitrary heavy-tailed noise with only bounded $(1 + \delta)$ moment ($\delta \in (0, 1)$), which is significant on its own strengths.

## 3  PRELIMINARIES

We will present the setting of LowHTR and introduce the common assumptions for theoretical analysis in this section. Denote $T$ as the total horizon, which may be unknown to the agent. At each round $t \in [T]$, the agent is given an arm set $\mathcal{X}_t \subseteq \mathbb{R}^{d_1 \times d_2}$ ($d_1 \asymp d_2$) that can be fixed or varying over time. Then the agent chooses an arm $X_t \in \mathcal{X}_t$ and observes the associated stochastic reward $y_t$ such that,

$$y_t = \langle X_t, \Theta^* \rangle + \eta_t, \tag{1}$$

where $\Theta^* \in \mathbb{R}^{d_1 \times d_2}$ is an unknown parameter matrix with rank $r \ll d_1 \wedge d_2$ and $\eta_t$ is the heavy-tailed noise. Specifically, we assume $\mathbb{E}(\eta_t | \mathcal{F}_t) = 0$ and $\mathbb{E}(|\eta_t|^{1+\delta} | \mathcal{F}_t) \le c$ for some $\delta \in (0, 1], c > 0$ conditional on the history filtration $\mathcal{F}_t = \{X_t, X_{t-1}, \eta_{t-1}, \dots, X_1, \eta_1\}$, which indicates that $\mathbb{E}(y_t | \mathcal{F}_t) = \langle X_t, \Theta^* \rangle$. The compact SVD of $\Theta^*$ can be written as $\Theta^* = UDV^\top$ for some $U \in \mathbb{R}^{d_1 \times r}$ and $V \in \mathbb{R}^{d_2 \times r}$,

and we denote $D_{ii}$ as its $i$-th largest singular value. Furthermore, we define $X_t^* := \arg\max_{X \in \mathcal{X}_t} \langle X, \Theta^* \rangle$ as the feature matrix of the optimal arm at round $t$, and the goal is to minimize the cumulative regret in total $T$ rounds formulated as $R_T = \sum_{t=1}^T \langle X_t^*, \Theta^* \rangle - \langle X_t, \Theta^* \rangle$.

Next, we present two mild and regular assumptions.

**Assumption 3.1.** We can find a sampling distribution $\mathcal{D}$ over $\mathcal{X}_t$ with the covariance matrix $\Sigma$, such that $\mathcal{D}$ is sub-Gaussian with parameter $\sigma^2 \asymp c_l := \lambda_{\min}(\Sigma) \asymp 1/(d_1 d_2)$.

Assumption 3.1 is commonly used in the modern low-rank matrix bandits [Lu et al., 2021, Kang et al., 2022], and can be easily satisfied in many cases. For instance, when $\mathcal{X}_t$ is a region in $\mathbb{R}^{d_1 \times d_2}$ (e.g., Euclidean unit ball), we can find such a sampling distribution if the convex hull of this region contains a ball with some constant radius. And when $\mathcal{X}_t$ is a finite set, it suffices if the arms are IID drawn from some sub-Gaussian distribution at each time. Note a random matrix $X \in \mathbb{R}^{d_1 \times d_2}$ follows sub-Gaussian distribution with parameter $\sigma^2$ if for any $t \in \mathbb{R}$ s.t.,

$$\mathbb{P}(\langle A, X \rangle \ge \sqrt{2} \|A\|_{\mathrm{F}} t) \le 2 \exp\left(-t^2/\sigma^2\right), \ \forall A \in \mathbb{R}^{d_1 \times d_2}.$$

**Assumption 3.2.** We have $\|\Theta^*\|_{\mathrm{F}} \le S$, and for any $t \in [T], X \in \mathcal{X}_t$, it holds that $\|X\|_{\mathrm{F}} \le S$.

Assumption 3.2 is very standard in contextual bandit literature. As a consequence, we can deduce that $\mathbb{E}(|y_t|^{1+\delta} | \mathcal{F}_t) \le 2^\delta S^2 + 2^\delta c := b$. Based on the conditions on the sub-Gaussian parameter $\sigma$ in Assumption 3.1, we can prove that $\|X\|_{\mathrm{F}}$ is bounded in a constant scale with high probability with its proof in Appendix G. But for simplicity and consistency with previous literature, we still impose this common assumption to bound $\|X\|_{\mathrm{F}}$ here. Note our work can be naturally extended to the generalized low-rank matrix bandit problem by further assuming the derivative of the inverse link function is bounded in the interval $[-S^2, S^2]$. Such an adaptation would result in the final regret bound being affected only by a constant factor, and we will leave it as our future work.

## 4  METHODS

In this section, we present our novel LowTO With Estimated Subspaces (LOTUS) algorithm for the LowHTR problem. Our algorithm runs in a batched format adapted from the doubling trick [Besson and Kaufmann, 2018]. And inspired by the success of the two-stage framework in ESTR Jun et al. [2019], in each batch our algorithm also first recovers the subspaces spanned by $\Theta^*$, and then invokes a new approach called LowTO that heavily penalizes on columns and rows complementary to our estimated subspaces. Contrasting prior works, our algorithm could dynamically switch between the exploration and exploitation stages so as to be

agnostic to the horizon $T$, which is significantly more useful. We further improve LOTUS to operate without knowing the sparsity $r$, which further enhances its practicality.

Initially, we will introduce the nuclear penalized Huber-type low-rank matrix estimator under heavy-tailed noise as follows. Contracting the results in [Yu et al., 2023], we further prove that our Huber-type estimator is robust to arbitrary heavy-tailed noise with the finite $(1 + \delta)$ moment for $\delta \in (0, 1)$ on the trace regression problem.

## 4.1 LOW-RANK MATRIX ESTIMATION

Suppose we collect $n$ pairs of data $\{(X_i, y_i)\}$ according to some distribution satisfying Assumption 3.1 for $X_i$ and the model of Eqn. (1) for the associated $y_i$ after time $n$. Define the Huber loss [Huber, 1965] $l_\tau(\cdot)$ parameterized by the robustification $\tau > 0$ [Sun et al., 2020] as:

$$l_\tau(x) = \begin{cases} x^2/2 & \text{if } |x| \leq \tau, \\ \tau|x| - \tau^2/2 & \text{if } |x| > \tau. \end{cases}$$

To obtain a low-rank matrix estimate, we use the nuclear norm penalization as a convex surrogate for the rank and implement the following nuclear norm regularized Huber regressor [Yu et al., 2023] to recover the subspaces under heavy-tailedness:

$$\widehat{\Theta} = \arg \min_{\Theta \in \mathbb{R}^{d_1 \times d_2}} \hat{L}_{\tau,[n]}(\Theta) + \lambda \|\Theta\|_{\text{nuc}}, \qquad (2)$$

$$\hat{L}_{\tau,[n]}(\Theta) = \frac{1}{n} \sum_{i \in [n]} l_\tau \left( y_i - \langle X_i, \Theta \rangle \right),$$

where $\tau$ and $\lambda$ stand for the Huber loss robustification and the nuclear norm penalization parameters, respectively.

We then establish the following statistical properties of the estimator defined in Eqn. (2):

**Theorem 4.1.** *By extending Assumption 3.1 with any order of $\sigma$ and $c_l$, With probability at least $1 - \epsilon$, the low-rank estimator $\widehat{\Theta}$ in Eqn. (2) with $\tau \asymp (n/(d + \ln(1/\epsilon)))^{\frac{1}{1+\delta}} c^{\frac{1}{1+\delta}}$ and $\lambda \asymp \sigma \left((d + \ln(1/\epsilon))/n\right)^{\frac{\delta}{1+\delta}} c^{\frac{1}{1+\delta}}$ satisfies*

$$\left\| \widehat{\Theta} - \Theta^* \right\|_F \leq C_1 \frac{\sigma}{c_l} \left( \frac{d + \ln(1/\epsilon)}{n} \right)^{\frac{\delta}{1+\delta}} c^{\frac{1}{1+\delta}} \sqrt{r},$$

*for some constant $C_1$ as long as we have $n \gtrsim dr\nu^3, d, \nu^2,$ and $(d - \ln(\epsilon))\sqrt{r\nu^3}$ with $\nu = \sigma^2/c_l$.*

The proof of Theorem 4.1 involves a construction of the restricted strong convexity for the empirical Huber loss function $\hat{L}_\tau(\cdot)$ and a deduction of an upper bound for $\left\| \nabla \hat{L}_\tau(\Theta^*) \right\|_{\text{op}}$, and the details are presented in Appendix A. Note Theorem 4.1 generally holds without any restriction on the scale of $\sigma$ and $c_l$. Provided the noise has a finite

variance, i.e., $\delta = 1$, the deduced $l_2$-error rate aligns with the minimax value [Fan et al., 2019] under the standard penalized low-rank estimator with sub-Gaussian noise. Based on our knowledge, this is the first error bound in the trace regression problem under noise with finite $(1 + \delta)$ moment $(\delta < 1)$ assuming nothing further.

To solve the convex optimization problem in Eqn. (2), we adopt the local adaptive majorize-minimization (LAMM) method [Fan et al., 2018, Sun et al., 2020, Yu et al., 2023] that is fast to use and scalable to large datasets. This method constructs an isotropic quadratic function to upper bound the Huber loss and utilizes a majorize-minimization algorithm for finding the optimal solution. One noteworthy advantage of this procedure is that the minimizer often yields a closed-form solution. Due to the space limit, we defer more details and the pseudocode to Appendix I.

## 4.2 LOTUS: THE RANK $r$ IS KNOWN

We will present our LOTUS algorithm in this subsection. To improve the two-stage framework introduced in Jun et al. [2019] which requires the knowledge of $T$ and to further yield robust performance against heavy-tailedness, our LO-TUS adaptively switches between exploration and exploitation in a batch manner without knowing $T$, and is equipped with a new LowTO algorithm designed for heavy-tailed rewards. The LOTUS algorithm is presented in Algorithm 1, with three core steps introduced in detail as follows:

**Adaptive Exploration and Exploitation:** Drawing inspiration from the doubling trick [Besson and Kaufmann, 2018], after some warm-up iterations of size $T_0$, our LOTUS operates with batches until termination where the batch sizes increase exponentially as $\{2^i\}_{i=1}^{+\infty}$. We define $\mathcal{H}_1$ and $\mathcal{H}_2$ as the history and exploration buffer index sets, where after time $t$ all the indexes $[t]$ of past observations are included in $\mathcal{H}_1$ while $\mathcal{H}_2$ only contains sample indexes particularly used for subspace estimation of $\Theta^*$. At the $i$-th batch of length $2^i$, we first set $T_1^i = \min\{(d^{2+4\delta}r^{1+\delta}2^{i+i\delta}/D_{rr}^{2+2\delta})^{\frac{1}{1+3\delta}}, 2^i\}$ as the exploration length, and we randomly sample $T_1$ arms according to the sampling distribution in Assumption 3.1 and put their indexes into both $\mathcal{H}_1$ and $\mathcal{H}_2$. Subsequently, we obtain an estimate $\widehat{\Theta}$ based on Eqn.(2) with samples indexed by $\mathcal{H}_2$, and then leverage the recovered subspaces in the remaining $T_2^i = 2^i - T_1^i$ rounds as the exploitation phase, where we invoke a new algorithm named LowTO. The details of this exploitation phase will be elaborated in the following two points. As shown in Algorithm 1 line 8, indexes of observations under LowTO are only added to $\mathcal{H}_1$ but not $\mathcal{H}_2$ and hence will not be used for matrix estimation. Unlike the traditional doubling trick that restarts the algorithm at each batch, our algorithm facilitates interaction across different batches. Specifically, at the $i$-th batch, it utilizes all the samples in $\mathcal{H}_1$ and $\mathcal{H}_2$ accumulated from the previous batches for more informed decision-making.

---

**Algorithm 1** LowTO With Estimated Subspaces (LOTUS)

---

**Input:** Arm set $\mathcal{X}_t$, sampling distribution $\mathcal{D}_t, \delta, T_0, \eta, \lambda, \{\lambda_{i,\perp}\}_{i=1}^{+\infty}$.
**Initialization:** The history buffer index set $\mathcal{H}_1 = \{\}$, the exploration buffer index set $\mathcal{H}_2 = \{\}$.

1: Pull arm $X_t \in \mathcal{X}_t$ according to $\mathcal{D}_t$ and observe payoff $y_t$. Then add $(X_t, y_t)$ into $\mathcal{H}_1$ and $\mathcal{H}_2$ for $t \leq T_0$.
2: **for** $i = 1, 2, \ldots$ until the end of iterations **do**
3:  Set the exploration length $T_1 = \min\left\{\left[\frac{d^{2+4\delta}r^{1+\delta}}{D_{rr}^{2+2\delta}}2^{i(1+\delta)}\right]^{\frac{1}{1+3\delta}}, 2^i\right\}$.
4:  For iteration $t$ from $|\mathcal{H}_1| + 1$ to $|\mathcal{H}_1| + T_1$, pull arm $X_t \in \mathcal{X}_t$ according to $\mathcal{D}_t$ and observe payoff $y_t$. Then add $(X_t, y_t)$ into $\mathcal{H}_1$ and $\mathcal{H}_2$
5:  Obtain the estimate $\widehat{\Theta}$ based on Eqn. (3) with $\mathcal{H}_2$, where we set $\tau_i \asymp \left(|\mathcal{H}_2|/(d + \ln(2^{i+1}/\epsilon))\right)^{\frac{1}{1+\delta}} c^{\frac{1}{1+\delta}}, \lambda_i \asymp \sigma\left((d + \ln(2^{i+1}/\epsilon))/|\mathcal{H}_2|\right)^{\frac{\delta}{1+\delta}} c^{\frac{1}{1+\delta}}$.
6:  Calculate the full SVD of $\widehat{\Theta} = [\widehat{U}, \widehat{U}_\perp]\widehat{D}[\widehat{V}, \widehat{V}_\perp]^\top$ where $\widehat{U} \in \mathbb{R}^{d_1 \times r}, \widehat{V} \in \mathbb{R}^{d_2 \times r}$.
7:  For $T_2 = 2^i - T_1$ rounds, invoke LowTO with $\delta, [\widehat{U}, \widehat{U}_\perp], [\widehat{V}, \widehat{V}_\perp], \lambda, \lambda_{i,\perp}, \mathcal{H}_1$ and obtain the updated $\mathcal{H}_1$.
8: **end for**

---

Another point to highlight is that our LOTUS algorithm can also be run in a more randomized manner with the same regret bound: at the $i$-th batch, there is an option to explore with a probability of $T_1^i/2^i$ and to exploit with the remaining probability. We defer its pseudocode to Appendix H. For simplicity, we consider our original approach in this work, which involves an initial exploration phase of deterministic length followed by the use of LowTO.

**Subspace Transformation:** At the $i$-th batch, after we randomly sample arms for a carefully designed duration and add their observations into $\mathcal{H}_2$, we first acquire the estimated $\widehat{\Theta}$ based on the current $\mathcal{H}_2$ as shown in Eqn. (3). With the knowledge of $r$, then we can obtain its corresponding full SVD as $\widehat{\Theta} = [\widehat{U}, \widehat{U}_\perp]\widehat{D}[\widehat{V}, \widehat{V}_\perp]^\top$ where $\widehat{U} \in \mathbb{R}^{d_1 \times r}, \widehat{U}_\perp \in \mathbb{R}^{d_1 \times (d_1-r)}, \widehat{V} \in \mathbb{R}^{d_2 \times r}$ and $\widehat{V}_\perp \in \mathbb{R}^{d_2 \times (d_2-r)}$.

$$\widehat{\Theta} = \arg\min_{\Theta \in \mathbb{R}^{d_1 \times d_2}} \hat{L}_{\tau_i, \mathcal{H}_2}(\Theta) + \lambda_i \|\Theta\|_{\text{nuc}} \quad (3)$$

Intuitively, Theorem 4.1 implies that our estimated column and row subspaces should align with the ground truth $U, V$. Borrowing the ideas from ESTR [Jun et al., 2019], we aim to transform the original LowHTR into the linear bandit problem under heavy-tailed rewards with some sparsity feature. Specifically, we first orthogonally rotate the actions set $\mathcal{X}_j$ in the exploitation phase as

$$\mathcal{X}_j^- = \left\{[\widehat{U}, \widehat{U}_\perp]^\top X [\widehat{V}, \widehat{V}_\perp] : X \in \mathcal{X}_j\right\}, \quad (4)$$

$$\Theta^{*,'} = [\widehat{U}, \widehat{U}_\perp]^\top \Theta^* [\widehat{V}, \widehat{V}_\perp]. \quad (5)$$

Define the total dimension $p := d_1 d_2$ and the effective dimension $k := p - (d_1 - r)(d_2 - r)$. We perform a tailored vectorization of the arm set $\mathcal{X}_j^-$ as in Algorithm 2 line 4 to obtain a new arm set $\mathcal{X}_t' \subseteq \mathbb{R}^p$, and denote $\theta^*$ to be the corresponding rearranged version of $\text{vec}(\Theta^{*,'})$ such that $\theta_{k+1:p}^* = \text{vec}(\Theta_{r+1:d_1, r+1:d_2}^{*,'})$. Then it holds that $\theta_{k+1:p}^*$ is nearly zero based on the results in Stewart [1990] and Theorem 4.1. The formal result is shown as follows for the

$i$-th batch with probability at least $1 - \epsilon$:

$$\|\theta_{k+1:p}^*\|_2 \lesssim S_\perp := \frac{r\sigma^2 c^{\frac{2}{1+\delta}}}{c_l^2 D_{rr}^2} \left(\frac{d + \ln(1/\epsilon)}{|\mathcal{H}_2|}\right)^{\frac{2\delta}{1+\delta}}, \quad (6)$$

with the parameter setting that

$$\tau_i \asymp (|\mathcal{H}_2|/(d + \ln(1/\epsilon)))^{\frac{1}{1+\delta}} c^{\frac{1}{1+\delta}},$$
$$\lambda_i \asymp \sigma\left((d + \ln(1/\epsilon))/|\mathcal{H}_2|\right)^{\frac{\delta}{1+\delta}} c^{\frac{1}{1+\delta}},$$

Its complete proof is presented in Appendix C. Consequently, we can simplify the LowHTR problem to an equivalent $p$-dimensional linear bandits under heavy-tailedness with a unique sparse pattern, i.e., the final $(p - k)$ entries of $\theta^*$ are almost zero based on Eqn. (6).

Following the recovery of row and column subspaces of $\Theta^*$ and the particular arm set transformation after $T_1^i$ rounds in the $i$-th batch, we will leverage the resulting almost-low-dimensional structure by using the following LowTO algorithm for the rest of the batch's duration.

**LowTO Algorithm:** To begin with, we reformulate the resulting $p$-dimensional linear bandit problem under heavy-tailed rewards in the following way: at round $t$, the agent chooses an arm $x_t \in \mathcal{X}_t'$ of dimension $p$ where $\mathcal{X}_t'$ is a rearranged vectorization of $\mathcal{X}_t^-$ as defined in Algorithm 2 line 4, and observes a noisy payoff $y_t = x_t^\top \theta^* + \eta_t$ mixed with some heavy-tailed noise $\eta_t$.

Our LowTO algorithm is presented in Algorithm 2. Inspired by LowOFUL in the ESTR method [Jun et al., 2019], to exploit the additional pattern of $\theta^*$ shown in Eqn. (6), we propose the almost-low-dimensional TOFU (LowTO) algorithm that extends the truncation-based TOFU [Shao et al., 2018]. The original TOFU trims the observed payoffs for each dimension individually and takes the contexts of historical arms into account for the truncation, which could yield a tight regret bound of order $\tilde{O}(pT^{\frac{1}{1+\delta}})$. As shown in Algorithm 2 line 2, our LowTO also truncates each entry of

**Algorithm 2** LowTO

**Input:** $T, \delta, [\widehat{U}, \widehat{U}_\perp], [\widehat{V}, \widehat{V}_\perp], \lambda_0, \lambda_\perp, \mathcal{H}_1$.

**Initialization:** $M = \sum_{(x,y)\in\mathcal{H}_1'} xx^\top + \Lambda = \sum_{t=1}^{|\mathcal{H}_1'|} x_{s,t} x_{s,t}^\top +$
$\Lambda, X^\top = [x_{s,1}, \ldots, x_{s,|\mathcal{H}_1'|}], [u_1, \ldots, u_p]^\top = M^{-\frac{1}{2}} X^\top$

with $\mathcal{H}_1' = \Big\{ \Big( x_{s,t}^\top = [\text{vec}(\widehat{U}^\top X \widehat{V})^\top, \text{vec}(\widehat{U}^\top X \widehat{V}_\perp)^\top,$
$\text{vec}(\widehat{U}_\perp^\top X \widehat{V})^\top, \text{vec}(\widehat{U}_\perp^\top X \widehat{V}_\perp)^\top], y_{s,t} = y \Big) : (X,y) \in \mathcal{H}_1 \Big\}.$
$\Lambda = \text{diag}([\underbrace{\lambda_0, \ldots, \lambda_0}_{k}, \underbrace{\lambda_\perp, \ldots, \lambda_\perp}_{p-k}])$

1: **for** $t = 1$ **to** $T$ **do**
2:   Get $\hat{y}_i = [y_{s,1}\mathbb{1}_{u_{i,1} y_{s,1} \leq b_{t-1}}, \ldots, y_{t-1}\mathbb{1}_{u_{i,|\mathcal{H}_1|+t-1} y_{t-1}}$
   $\leq b_{t-1}]^\top$ for $i \in [p]$, where $\hat{y}_i \in \mathbb{R}^{|\mathcal{H}_1|+t-1}$.
3:   Calculate $\hat{\theta}_{t-1} = M^{-1/2}[u_1^\top \hat{y}_1, \ldots, u_p^\top \hat{y}_p]^\top$.
4:   Transform the arm set $\mathcal{X}_t$ as

$\mathcal{X}_t' = \Big\{ [\text{vec}(\widehat{U}^\top X \widehat{V})^\top, \text{vec}(\widehat{U}^\top X \widehat{V}_\perp)^\top, \text{vec}(\widehat{U}_\perp^\top X \widehat{V})^\top,$
$\text{vec}(\widehat{U}_\perp^\top X \widehat{V}_\perp)^\top]^\top \in \mathbb{R}^p : X \in \mathcal{X}_t \Big\}.$

5:   Pull $x_t = \arg\max_{x\in\mathcal{X}_t'} x^\top \hat{\theta}_{t-1} + \beta_{t-1} \|x\|_{M^{-1}}$
   and observe the reward $y_t$.
6:   Restore $x_t$ into its original matrix form $X_t$ and then
   add $(X_t, y_t)$ into $\mathcal{H}_1$.
7:   Update $M = M + x_t x_t^\top$, $X^\top = [X^\top, x_t]$ and
   $[u_1, \ldots, u_p]^\top = M^{-1/2} X^\top$.
8: **end for**
9: **return** The history buffer $\mathcal{H}_1$.

$M^{-1/2} x_i y_i$ for $i = 1, \ldots, t-1$ at time $t$ by some increasing threshold $b_t$, Different from TOFU, when calculating the estimator $\hat{\theta}$ in Algorithm 2 line 3, we put a weighted regularizer as the diagonal matrix $\Lambda = \text{diag}(\lambda, \ldots, \lambda, \lambda_\perp, \ldots, \lambda_\perp)$ with $\lambda$ only applied to the first $k$ coordinates. By amplifying $\lambda_\perp$, we ensure greater penalization is applied to the final $p-k$ elements of $\hat{\theta}$ leading to their diminished values, and this phenomenon is well intended under the almost-low-dimensional structure. Subsequently, we utilize a UCB-based criterion to choose the pulled arm according to Algorithm 2 line 5, where we also decrease the variation of the last $p-k$ elements with $M^{-1}$ to further reduce their impact on the decision-making. It is also noteworthy that we always reuse all the past observations stored in $\mathcal{H}_1$ at each batch when initializing the matrix $M$, which can facilitate a consistent and accurate estimator $\hat{\theta}$ in the early stage of the exploitation phase. And the randomly drawn samples in $\mathcal{H}_1$ contain more stochasticity and thus are more preferable for the parameter estimation.

We then state the regret bound of LowTO in Theorem 4.2:

**Theorem 4.2.** *Suppose the input $\mathcal{H}_1$ is of size $H \lesssim$*

$T$ *and we run our LowTO algorithm for $T$ rounds. By setting* $b_t = (b/\log(2p/\epsilon))^{\frac{1}{1+\delta}}(t+H)^{\frac{1-\delta}{2+2\delta}}, \beta_t = 4\sqrt{p}b^{\frac{1}{1+\delta}}\log(2p/\epsilon)^{\frac{\delta}{1+\delta}}(t+H)^{\frac{1-\delta}{2+2\delta}} + \sqrt{\lambda_0}S + \sqrt{\lambda_\perp}S_\perp$ *with* $\lambda_\perp = S^2 T_2 / (k\log(1 + \frac{S^2 T}{k\lambda_0}))$, *with probability at least $1 - \epsilon$, the regret of LowTO can be bounded by:*

$$\tilde{O}\left( \sqrt{kp}\,(T+H)^{\frac{1}{1+\delta}} + \sqrt{kT} + S_\perp T \right),$$

*where $S_\perp$ is the upper bound of $\|\theta_{k+1:p}\|_2$ as shown in Eqn. (6) depending on $|\mathcal{H}_2|$.*

In standard linear bandit under heavy-tailed noise case, we can recover the same regret bound of TOFU in the order of $\tilde{O}(p \cdot T^{\frac{1}{1+\delta}})$ by setting $S_\perp = S$ and $\lambda_\perp = \lambda$.

**Overall regret:** Now we are ready to present the overall regret bound for LOTUS in the following Theorem 4.3.

**Theorem 4.3.** *By using the configuration of LowTO described in Theorem 4.2 and the parameter values of LOTUS shown in Algorithm 1 for each batch, and set $\epsilon$ as $\epsilon/2^{i+1}$ in $\beta_t$ (formulated in Theorem 4.2) for the $i$-th batch. Then with probability at least $1 - \epsilon$, it holds that*

$$R(T) \leq \tilde{O}\left( d^{\frac{2+4\delta}{1+3\delta}} r^{\frac{1+\delta}{1+3\delta}} T^{\frac{1+\delta}{1+3\delta}} / D_{rr}^{\frac{2+2\delta}{1+3\delta}} + d^{\frac{3}{2}} r^{\frac{1}{2}} T^{\frac{1}{1+\delta}} \right),$$

*under the condition that $T_1 \geq 5d^{\frac{1+2\delta}{\delta}} r^{\frac{1+\delta}{2\delta}} / D_{rr}^{\frac{1+\delta}{\delta}}$. Furthermore, we can simplify the above result as*

$$R(T) \leq \begin{cases} \tilde{O}\left( d^{\frac{3}{2}} r^{\frac{1}{2}} T^{\frac{1}{2}} / D_{rr} \right), \delta = 1; \\ \tilde{O}\left( d^{\frac{3}{2}} r^{\frac{1}{2}} T^{\frac{1}{1+\delta}} \right), \delta < 1, T \gtrsim (dr)^{\frac{1+\delta}{2\delta}} / D_{rr}^{\frac{2(1+\delta)^2}{\delta(1-\delta)}}. \end{cases}$$

Note the regret bound in Theorem 4.3 improves upon the one attained for a simple linear bandit reduction, which contains the order of $d^2$. When the rewards have bounded variance, i.e., $\delta = 1$, our regret bound matches the modern one for low-rank matrix bandit under sub-Gaussian noise up to logarithmic terms [Lu et al., 2021, Kang et al., 2022].

### 4.3 LOTUS: THE RANK $r$ IS UNKNOWN

While all existing algorithms for low-rank matrix bandits require prior knowledge of the rank $r$, this information is never revealed to agents in real-world applications, and hence misspecification of $r$ will not only undermine the theoretical foundations but also severely compromise the performance of these methods. To solve this crucial challenge, in this section we aim to enhance our LOTUS algorithm to be agnostic to $r$ even under the more complex heavy-tailed scenario. For the Lasso bandit, which is another popular and easier high-dimensional bandit with sparsity, some algorithms [Oh et al., 2021, Ariu et al., 2022] free of the sparsity

index have been recently introduced. However, when compared with our work, all of them necessitate some additional assumptions on the structure of the underlying parameter as well as the sampling distribution. For example, Oh et al. [2021] further assumes that the active entries of the parameter vector are relatively independent and the skewness of the sampling distribution is bounded. This fact substantiates the huge difficulty of devising an efficient algorithm for LowHRT without additional conditions. Note our work also opens up a potential avenue for exploring low-rank matrix bandits without the need for knowledge about $r$, and we believe that completely addressing this intriguing problem must require more specific assumptions and investigations.

To improve our batched-explore-then-exploit-based LOTUS algorithm, an intuitive idea is to estimate the effective rank of $\widehat{\Theta}$ right after the matrix recovery in each batch. By trimming the estimated singular values $\{D_{ii}\}_{i=1}^d$ with some craftily designed increasing sequence that is deduced from Theorem 4.1, we could obtain a useful rank $\hat{r}$ with $\hat{r} \leq r$ and then only focus on the top-$\hat{r}$ row and column subspaces. We can demonstrate that all the ground truth singular values $\{D_{ii}\}_{i=\hat{r}+1}^d$ omitted are nearly null and hence negligible. Therefore, by penalizing the subspaces parallel to those omitted directions with a similar idea used in our original LOTUS, we could enjoy the low-rank benefit of LowHRT. Specifically, to modify line 6 and line 7 in Algorithm 1, we abuse the notation here and denote $\widehat{D}$ as the singular value matrix of $\widehat{\Theta}$ that is deduced in line 5. Subsequently, we estimate the useful rank $\hat{r}$ as

$$\hat{r} = \min \left\{ i \in [d+1] : \widehat{D}_{ii} \leq C_1 \frac{\sigma \sqrt{i}}{c_l} \left( \frac{d + \ln\left(2^{i+1}/\epsilon\right)}{|\mathcal{H}_2|} \right)^{\frac{\delta}{1+\delta}} \right.$$
$$\left. \cdot c^{\frac{1}{1+\delta}} \right\} - 1 \wedge 1,$$

where $C_1$ is some specific constant in Theorem 4.1 and $\widehat{D}_{(d+1)(d+1)}$ is set to be 0 to avoid the empty set case. Afterward, we rewrite the full SVD of $\widehat{\Theta}$ as $\widehat{\Theta} = [\widehat{U}, \widehat{U}_\perp] \, \widehat{D} \, [\widehat{V}, \widehat{V}_\perp]^\top$ with $\widehat{U} \in \mathbb{R}^{d_1 \times \hat{r}}, \widehat{V} \in \mathbb{R}^{d_2 \times \hat{r}}$ for each batch in line 6. In new line 7 of our improved LOTUS, we then input the new $[\widehat{U}, \widehat{U}_\perp]$ and $[\widehat{V}, \widehat{V}_\perp]$ with the estimated rank $\hat{r}$ as described above, and the effective dimension $k$ in the following subspace estimation and LowTO implementation will become $k = p - (d_1 - \hat{r})(d_2 - \hat{r})$. Note $\hat{r}$ might differ across different batches, but $\hat{r} \leq r$ consistently holds.

Conclusively, we can obtain the following regret bound of our improved LOTUS algorithm agnostic to $r$:

**Theorem 4.4.** *By using the same setting and conditions of LOTUS as described in Theorem 4.3 and Algorithm 1 with $T_1 = \min \left\{ d \cdot 2^{\frac{i(1+\delta)}{1+2\delta}}, 2^i \right\}$ in line 3 of Algorithm 1, and utilizing the estimated useful rank $\hat{r}$ to set the corresponding value of $k$ at each batch, the cumulative regret of our LOTUS agnostic to $r$ can be bounded as*

$$R(T) \leq \tilde{O} \left( d^{\frac{3}{2}} r^{\frac{1}{2}} T^{\frac{1}{1+\delta}} + d r^{\frac{3}{2}} T^{\frac{1+\delta}{1+2\delta}} \right),$$

*with probability at least $1 - \epsilon$.*

While there exists a disparity between our derived regret bound in cases where $r$ remains undisclosed and the optimal one, as previously discussed in this section, it would prove exceptionally difficult to devise an algorithm for LowHTR that remains agnostic to $r$ while achieving a similar regret bound without more stringent assumptions. Solving this issue would necessitate the formulation of more specific assumptions on the underlying structure of the arm matrices and $\Theta^*$.

Moreover, we will showcase the superior efficiency of our LOTUS algorithm in both scenarios, whether the agent possesses knowledge of $r$ or not, in the following experimental results in Section 6.

## 5 LOWER BOUNDS

In this section, we provide a lower bound for the expected cumulative regret in LowHTR particularly regarding the order of $T$. The result is given as follows:

**Theorem 5.1.** *Under the LowHTR problem with $d, r, T$ and $S = 1$ in Assumption 3.2, there exists an instance with a fixed $\mathcal{X}_t$ containing $(d-1)r$ arms for which any algorithm must suffer an expected regret of order $\Omega(d^{\frac{\delta}{1+\delta}} r^{\frac{\delta}{1+\delta}} T^{\frac{1}{1+\delta}})$, i.e., $\mathbb{E}(R_T) \gtrsim d^{\frac{\delta}{1+\delta}} r^{\frac{\delta}{1+\delta}} T^{\frac{1}{1+\delta}} \gtrsim T^{\frac{1}{1+\delta}}$.*

Theorem 5.1 demonstrates that our LOTUS could attain the lower bound for LowHTR regarding the order of $T$ when $r$ is given. And this lower bound is tight with $r = d$ and finite arm sets since it matches the minimax rate for standard linear bandits under heavy-tailed noise [Xue et al., 2020]. Further exploring the regret lower bound for $d$ and $r$ under LowHTR is notably challenging, given the fact that even the simpler low-rank matrix bandits under sub-Gaussian noise this problem is not thoroughly studied [Kang et al., 2022]. And the regret lower bound may differ in the order of $d$ when the arm set is infinitely large and arbitrary [Shao et al., 2018]. We will leave them as future directions.

## 6 EXPERIMENTS

We demonstrate that our proposed LOTUS yields superior performance over the existing LowESTR algorithm [Lu et al., 2021] in the presence of heavy-tailed noise under a suite of simulations. Since our work is the first one to study the LowHTR problem and currently there is no existing method for comparison, we utilize the LowESTR algorithm specifically designed for the sub-Gaussian noise to validate the robustness of our proposed LOTUS. LowESTR also borrows the idea of the two-stage framework from ESTR, and it improves upon ESTR on the computational efficiency of the matrix recovery step. It requires both the knowledge

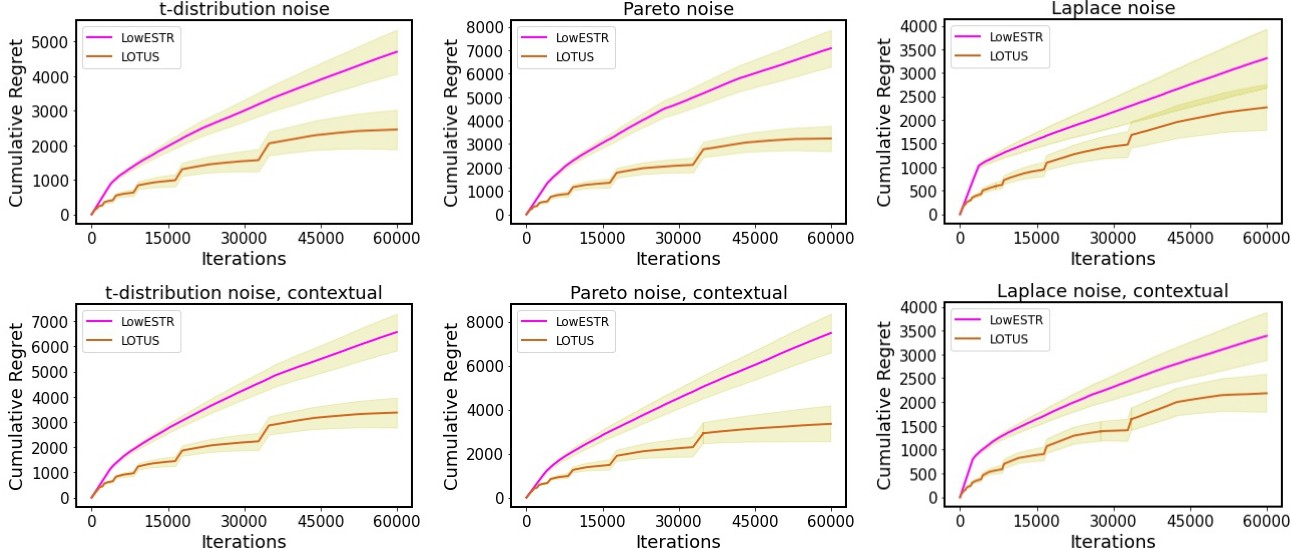

Figure 1: Plots of cumulative regrets of LowESTR and our proposed LOTUS with fixed or changing contextual arm set under t-distribution, Pareto, and Laplace heavy-tailed noise. We use the LOTUS algorithm agnostic to $r$ in the first three experiments displayed in the first row, and we utilize the value of $r$ in LOTUS in experiments shown in the second row.

of the horizon $T$ and the rank $r$ as inputs. In the following experiments, we showcase that it becomes vulnerable and achieves suboptimal performance under heavy-tailed noise in practice as expected. The values of hyperparameters in our LOTUS are strictly aligned with their theoretical results deduced in Theorem 4.1 and Theorem 4.2.

We consider two different settings of the parameter matrices $\Theta^*$ with $d_1 = d_2 = 10$ and $r = 2$. For the first scenario, we set the parameter matrix as a diagonal matrix $\Theta^* = \text{diag}([7, 4, 0, \ldots, 0])$. The arm set is fixed where we draw 500 random matrices from $\{X \in \mathbb{R}^{10 \times 10} : \|X\|_{\mathrm{F}} \leq 1\}$ in the beginning. And we implement the improved LOTUS algorithm introduced in Subsection 4.3 that is unaware of the rank $r$ in this scenario. For the second case, we consider a more challenging parameter matrix $\Theta^*$ such that its first row represents a random vector of norm 7 and its second row is a perpendicular vector of norm 4 with other entries set to 0. Contrasting the first scenario, we consider a contextual arm set with 10 feature matrices drawn from $\{X \in \mathbb{R}^{10 \times 10} : \|X\|_{\mathrm{F}} \leq 1\}$ at each round. And we use the original LOTUS algorithm introduced in Subsection 4.2 requiring the knowledge of $r = 2$. For the heavy-tailed noise $\eta_t$, we consider the following three types of distribution for both scenarios introduced above:

- **Student's t-distribution:** The density function is given as $f(x) \asymp (1 + x^2/\nu)^{-\frac{\nu+1}{2}}$ with degree of freedom parameter $\nu > 0$ and $x \in \mathbb{R}$. By setting $\nu = 1.7$, it has infinite variance but finite 1.5 moment bounded by 6. The heavy-tail index is equal to 1.60.[1]

- **Pareto distribution:** The density function is given as $f(x) \asymp \alpha/(x + 1)^{\alpha+1}$ for some shape parameter $\alpha > 0$ and $x > 0$. By setting $\alpha = 1.9$, it also has infinite variance but finite 1.5 moment bounded by 5. And the heavy-tail index is equal to 2.20.

- **Laplace distribution:** The density distribution is formulated as $f(x) \asymp \exp(-|x|/b)$ with some scale parameter $b$ for $x \in \mathbb{R}$. By setting $b = 1$, the distribution possesses a finite variance bounded by 2. The heavy-tail index of this distribution is 1.36.

According to Figure 1, we observe that our LOTUS algorithm consistently exhibits superior and more resilient performance across all six scenarios compared to LowESTR. This advantage is particularly evident when dealing with distributions with a higher heavy-tail index, which is aligned with our expectations. On the contrary, LowESTR performs fairly in the presence of Laplace noise with a finite variance but struggles when faced with Pareto noise possessing stronger heavy-tailedness. Furthermore, it is noteworthy that the cumulative regret of the LOTUS algorithm exhibits a batch-wise increase, with a progressively clearer sub-linear pattern emerging in subsequent batches. This fact firmly validates the practical superiority of our LOTUS algorithm under both cases when the rank $r$ is presented or not.

# 7 CONCLUSIONS

In this work, we introduce and examine the new problem of LowHTR, and we propose a robust algorithm named LO-

---

[1]A greater heavy-tail index [Hoaglin et al., 2000] above 1 indi- cates stronger fluctuation and heavy-tailedness of the distribution.

TUS that can be agnostic to $T$ and even the rank $r$ with a slightly milder regret bound. We also develop a matching lower bound to demonstrate our LOTUS is nearly optimal in the order of $T$. Meanwhile, we prove that our Huber-type estimator could solve the trace regression problem under arbitrary heavy-tailed noise with finite $(1 + \delta)$ moment ($\delta \in (0, 1]$) and its Frobenious norm error is of scale $\tilde{O}((d/n)^{\frac{\delta}{1+\delta}} \mathbb{E}(|\eta|^{1+\delta})^{\frac{1}{1+\delta}})$. The practical superiority of our proposed method is validated under simulations.

**Limitations:** Although our work represents the first solution to the low-rank matrix bandits without knowing $r$, it leaves a gap with our deduced lower bound. Closing this regret gap seems highly non-trivial without additional assumptions [Oh et al., 2021], and we will leave it as a future work.

## Acknowledgements

We appreciate the constructive feedback from the anonymous reviewers and area chair. This work was partially supported by the National Science Foundation under grants CCF-1934568, DMS-1916125, DMS-2113605, DMS-2210388, IIS-2008173 and IIS2048280.

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

# Appendix for Low-rank Matrix Bandits with Heavy-tailed Rewards (Supplementary Material)

**Yue Kang**[1]  **Cho-Jui Hsieh**[2]  **Thomas C. M. Lee**[1]

[1]Department of Statistics, UC Davis, Davis, CA, USA
[2]Google and Department of Computer Science, UCLA, Los Angeles, CA, USA

## A ANALYSIS OF THEOREM 4.1

The analysis of Theorem 4.1 is inspired by and extended from Yu et al. [2023].

### A.1 PRELIMINARIES

**Lemma A.1.** *(Bernstein Inequality) Let $X$ be a random variable with mean $\mu$ and variance $\sigma^2$. Assume we can find some $b > 0$ such that*

$$\mathbb{E}|X - \mu|^k \leq \frac{1}{2}k!\sigma^2 b^{k-2}, \ k = 3, 4, 5, \ldots$$

*Then it holds that*

$$\mathbb{P}\left(|X - \mu| \geq t\right) \leq 2\exp\left(-\frac{t^2}{2(\sigma^2 + bt)}\right), \ \forall t > 0.$$

**Corollary A.2.** *(Adapted from Bernstein Inequality) Let $X$ be a random variable with mean $\mu$ and variance $\sigma^2$. Assume we can find some $b > 0$ such that*

$$\mathbb{E}|X - \mu|^k \leq \frac{1}{2}k!\sigma^2 b^{k-2}, \ k = 3, 4, 5, \ldots$$

*Then it holds that*

$$\mathbb{P}\left(X - \mu \geq \sqrt{2t}\sigma + 2bt\right) \leq \exp\left(-t\right), \ \forall t > 0.$$

*Proof.* Based on Lemma A.1, we have that for any $t > 0$

$$\mathbb{P}\left(X - \mu \geq \sqrt{2t}\sigma + 2bt\right) \leq \exp\left(-\frac{(\sqrt{2t}\sigma + 2bt)^2}{2\sigma^2 + 2b(\sqrt{2t}\sigma + 2bt)}\right) \leq \exp\left(-\frac{2\sigma^2 t + 4b^2 t^2 + 4\sqrt{2}b\sigma t^{\frac{3}{2}}}{2\sigma^2 + 4\sigma^2 t + 2\sqrt{2}b\sigma\sqrt{t}}\right)$$
$$\leq \exp(-t).$$

$\square$

**Definition A.3.** (Local Restricted Strong Convexity) For the empirical loss function $\hat{L}_\tau(\cdot)$, we can define the event of local restricted strong convexity $\mathcal{E}(s, l, \kappa)$ in terms of the radius parameter $s, l$ and the curvature parameter $\kappa$ as

$$\mathcal{E}(s, l, \kappa) = \left\{\inf_{\Theta \in \mathcal{M}(\Theta^*, s, l)} \frac{\langle \nabla\hat{L}_\tau(\Theta) - \nabla\hat{L}_\tau(\Theta^*), \Theta - \Theta^* \rangle}{\|\Theta - \Theta^*\|_{\mathrm{F}}^2} \geq \kappa\right\},$$

where $\mathcal{M}(\Theta^*, s, l) = \left\{\Theta \in \mathbb{R}^{d_1 \times d_2} : \|\Theta - \Theta^*\|_{\mathrm{F}} \leq s, \|\Theta - \Theta^*\|_{\mathrm{nuc}} \leq l\|\Theta - \Theta^*\|_{\mathrm{F}}\right\}$.

We assume $d_1 \geq d_2$ without loss of generality, and denote $\widehat{\Delta} := \widehat{\Theta} - \Theta^*$ in the following argument. To start with, we will show that our target $\left\|\widehat{\Delta}\right\|_F$ can be bounded conditioned on the event $\mathcal{E}(s, l, \kappa)$ and $\lambda \geq 2 \left\|\nabla \hat{L}_\tau(\Theta^*)\right\|_{\text{op}}$.

**Theorem A.4.** *Conditioned on the event $\lambda \geq 2 \left\|\nabla \hat{L}_\tau(\Theta^*)\right\|_{\text{op}}$ and the event $\mathcal{E}(s, l, \kappa)$ with $s \geq 9\sqrt{r}\frac{\lambda}{\kappa}$ and $l \geq 4\sqrt{2r}$, then we can deduce that*

$$\left\|\widehat{\Delta}\right\|_F = \left\|\widehat{\Theta} - \Theta^*\right\|_F \leq 9\sqrt{r} \cdot \frac{\lambda}{\kappa}.$$

*Proof.* We will prove Theorem A.4 by contradiction. Assume we have that $\lambda \geq 2 \left\|\nabla \hat{L}_\tau(\Theta^*)\right\|_{\text{op}}$ and $\mathcal{E}(s, l, \kappa)$ holds with $s \geq 9\sqrt{r}\frac{\lambda}{\kappa}$ and $l \geq 4\sqrt{2r}$, and we assume $\left\|\widehat{\Delta}\right\|_F > 9\sqrt{r} \cdot \frac{\lambda}{\kappa}$ holds. Define $\widetilde{\Theta}_x = \Theta^* + x(\widehat{\Theta} - \Theta^*)$ as a function of $x \in [0, 1]$, then there exists some $\zeta \in (0, 1)$ such that $\widetilde{\Theta}_\zeta = \Theta^* + \zeta(\widehat{\Theta} - \Theta^*)$ satisfying $\left\|\widetilde{\Theta}_\zeta - \Theta^*\right\|_F = 9\sqrt{r} \cdot \frac{\lambda}{\kappa}$ since $\left\|\widetilde{\Theta}_x - \Theta^*\right\|_F$ is a continuous function in terms of $x \in [0, 1]$. Furthermore, we define $Q(x) = \hat{L}_\tau(\widetilde{\Theta}_x) - \hat{L}_\tau(\Theta^*) - \langle \nabla \hat{L}_\tau(\Theta^*), \widetilde{\Theta}_x - \Theta^* \rangle$. Note $x \in [0, 1] \to Q(x)$ can be easily shown as a convex function: first, we observe that $\widetilde{\Theta}_x$ is a linear function of $x$, and the Huber loss function defined in Section 4.1 is convex [Huber, 1965], which implies that $\hat{L}_\tau(\widetilde{\Theta}_x)$ is convex. On the other hand, the inner product $\langle \nabla \hat{L}_\tau(\Theta^*), \widetilde{\Theta}_x - \Theta^* \rangle$ is bi-linear and hence naturally convex as well. Therefore, we know that $Q'(x) = \langle \nabla \hat{L}_\tau(\widetilde{\Theta}_x) - \nabla \hat{L}_\tau(\Theta^*), \widehat{\Theta} - \Theta^* \rangle$ is monotonically increasing. And it holds that

$$\zeta Q'(\zeta) \leq \zeta Q'(1) \implies \langle \nabla \hat{L}_\tau(\widetilde{\Theta}_\zeta) - \nabla \hat{L}_\tau(\Theta^*), \widetilde{\Theta}_\zeta - \Theta^* \rangle \leq \zeta \langle \nabla \hat{L}_\tau(\widehat{\Theta}) - \nabla \hat{L}_\tau(\Theta^*), \widehat{\Theta} - \Theta^* \rangle \tag{7}$$

To bound the right-hand side of Eqn. (7), since $\widehat{\Theta}$ is the solution to the convex optimization problem in Eqn. (2), then we have the sub-gradient condition as:

$$\langle \nabla \hat{L}_\tau(\widehat{\Theta}) + \lambda \widehat{Z}, \widehat{\Theta} - \Theta^* \rangle \leq 0, \quad \text{where } \widehat{Z} \in \partial \left\|\widehat{\Theta}\right\|_{\text{nuc}}.$$

Due to the definition of the sub-gradient, it holds that $\|\Theta^*\|_{\text{nuc}} \geq \|\widehat{\Theta}\|_{\text{nuc}} + \langle \widehat{Z}, \Theta^* - \widehat{\Theta} \rangle$. By assuming $\lambda \geq 2 \left\|\nabla \hat{L}_\tau(\Theta^*)\right\|_{\text{op}}$, we can have that

$$\langle \nabla \hat{L}_\tau(\widehat{\Theta}) - \nabla \hat{L}_\tau(\Theta^*), \widehat{\Theta} - \Theta^* \rangle \leq \langle \lambda \widehat{Z}, \Theta^* - \widehat{\Theta} \rangle + \langle \nabla \hat{L}_\tau(\Theta^*), \Theta^* - \widehat{\Theta} \rangle$$
$$\leq \lambda \left( \|\Theta^*\|_{\text{nuc}} - \|\widehat{\Theta}\|_{\text{nuc}} \right) + \frac{\lambda}{2} \left\|\Theta^* - \widehat{\Theta}\right\|_{\text{nuc}} \leq \frac{3\lambda}{2} \left\|\widehat{\Delta}\right\|_{\text{nuc}}$$

To bound $\left\|\widehat{\Delta}\right\|_{\text{nuc}}$, we utilize the regular procedure [Negahban and Wainwright, 2011, Yu et al., 2023]. We restate the notation and define the reduced SVD of $\Theta^*$ as $\Theta^* = U\Sigma V^\top$ with $U \in \mathbb{R}^{d_1 \times r}$ and $V \in \mathbb{R}^{d_2 \times r}$. Then we denote two sets as:

$$\mathbb{M} = \left\{ \Theta \in \mathbb{R}^{d_1 \times d_2} : \text{row}(\Theta) \subseteq \text{col}(V), \text{col}(\Theta) \subseteq \text{col}(U) \right\},$$
$$\bar{\mathbb{M}}^\perp = \left\{ \Theta \in \mathbb{R}^{d_1 \times d_2} : \text{row}(\Theta) \subseteq \text{col}(V)^\perp, \text{col}(\Theta) \subseteq \text{col}(U)^\perp \right\},$$

and hence $\mathbb{M} \subseteq \bar{\mathbb{M}}$. Next we will show that $\|\widehat{\Delta}_{\bar{\mathbb{M}}^\perp}\|_{\text{nuc}} \leq 3\|\widehat{\Delta}_{\bar{\mathbb{M}}}\|_{\text{nuc}}$ in the following part. First, since $\widehat{\Theta}$ is the solution to the problem defined in Eqn.(2), we have that

$$\hat{L}_\tau(\widehat{\Theta}) + \lambda \left\|\widehat{\Theta}\right\|_{\text{nuc}} \leq \hat{L}_\tau(\Theta^*) + \lambda \|\Theta^*\|_{\text{nuc}} \iff \hat{L}_\tau(\widehat{\Theta}) - \hat{L}_\tau(\Theta^*) \leq \lambda \left( \|\Theta^*\|_{\text{nuc}} - \left\|\widehat{\Theta}\right\|_{\text{nuc}} \right).$$

For the left-hand side, it holds that

$$\hat{L}_\tau(\widehat{\Theta}) - \hat{L}_\tau(\Theta^*) \geq \langle \nabla \hat{L}_\tau(\Theta^*), \widehat{\Theta} - \Theta^* \rangle \geq - \left\|\nabla \hat{L}_\tau(\Theta^*)\right\|_{\text{op}} \left\|\widehat{\Delta}\right\|_{\text{nuc}}$$
$$\geq - \left\|\nabla \hat{L}_\tau(\Theta^*)\right\|_{\text{op}} \left( \left\|\widehat{\Delta}_{\mathbb{M}}\right\|_{\text{nuc}} + \left\|\widehat{\Delta}_{\bar{\mathbb{M}}^\perp}\right\|_{\text{nuc}} \right) \geq -\frac{\lambda}{2} \left( \left\|\widehat{\Delta}_{\bar{\mathbb{M}}}\right\|_{\text{nuc}} + \left\|\widehat{\Delta}_{\bar{\mathbb{M}}^\perp}\right\|_{\text{nuc}} \right). \tag{8}$$

And for the right-hand side, we have that

$$\left\|\widehat{\Theta}\right\|_{\text{nuc}} = \left\|\Theta^* + \widehat{\Delta}\right\|_{\text{nuc}} = \left\|\Theta_{\mathbb{M}}^* + \widehat{\Delta}_{\bar{\mathbb{M}}} + \widehat{\Delta}_{\bar{\mathbb{M}}^\perp}\right\|_{\text{nuc}} \geq \|\Theta_{\mathbb{M}}^*\|_{\text{nuc}} + \left\|\widehat{\Delta}_{\bar{\mathbb{M}}^\perp}\right\|_{\text{nuc}} - \left\|\widehat{\Delta}_{\bar{\mathbb{M}}}\right\|_{\text{nuc}},$$

and hence we have that

$$\|\Theta^*\|_{\text{nuc}} - \left\|\widehat{\Theta}\right\|_{\text{nuc}} = \|\Theta^*_{\check{\mathbb{M}}}\|_{\text{nuc}} - \left\|\widehat{\Theta}\right\|_{\text{nuc}} \leq \left\|\widehat{\Delta}_{\check{\mathbb{M}}}\right\|_{\text{nuc}} - \left\|\widehat{\Delta}_{\check{\mathbb{M}}^\perp}\right\|_{\text{nuc}}. \tag{9}$$

Combining the results from Eqn. (8) and Eqn. (9), we can deduce that $\|\widehat{\Delta}_{\check{\mathbb{M}}^\perp}\|_{\text{nuc}} \leq 3\|\widehat{\Delta}_{\check{\mathbb{M}}}\|_{\text{nuc}}$. Next, since we have that $\text{rank}(\widehat{\Delta}_{\check{\mathbb{M}}}) \leq 2r$, then based on Cauchy-Schwarz inequality it holds that

$$\left\|\widehat{\Delta}\right\|_{\text{nuc}} \leq \left\|\widehat{\Delta}_{\mathbb{M}}\right\|_{\text{nuc}} + \left\|\widehat{\Delta}_{\check{\mathbb{M}}^\perp}\right\|_{\text{nuc}} \leq \left\|\widehat{\Delta}_{\check{\mathbb{M}}}\right\|_{\text{nuc}} + \left\|\widehat{\Delta}_{\check{\mathbb{M}}^\perp}\right\|_{\text{nuc}} \leq 4\left\|\widehat{\Delta}_{\check{\mathbb{M}}}\right\|_{\text{nuc}} \leq 4\sqrt{2r}\left\|\widehat{\Delta}_{\check{\mathbb{M}}}\right\|_{\text{F}} \leq 4\sqrt{2r}\left\|\widehat{\Delta}\right\|_{\text{F}}.$$

Therefore, we can show that $\left\|\widetilde{\Theta}_\zeta - \Theta^*\right\|_{\text{nuc}} \leq 4\sqrt{2r}\left\|\widetilde{\Theta}_\zeta - \Theta^*\right\|_{\text{F}}$. And remember that we assume $\left\|\widetilde{\Theta}_\zeta - \Theta^*\right\|_{\text{F}} = 9\sqrt{r} \cdot \frac{\lambda}{\kappa}$. These facts indicate that $\widetilde{\Theta}_\zeta \in \mathcal{M}(\Theta^*, s, l)$ with $s \geq 9\sqrt{r}\frac{\lambda}{\kappa}$ and $l \geq 4\sqrt{2r}$. Therefore, based on the local restricted strong convexity, we have

$$\kappa\zeta\left\|\widehat{\Delta}\right\|_{\text{F}}\left\|\widetilde{\Theta}_\zeta - \Theta^*\right\|_{\text{F}} = \kappa\left\|\widetilde{\Theta}_\zeta - \Theta^*\right\|_{\text{F}}^2 \leq \langle\nabla\hat{L}_\tau(\widetilde{\Theta}_\zeta) - \nabla\hat{L}_\tau(\Theta^*), \widetilde{\Theta}_\zeta - \Theta^*\rangle.$$

For the left-hand side, it holds that

$$\kappa\zeta\left\|\widehat{\Delta}\right\|_{\text{F}}\left\|\widetilde{\Theta}_\zeta - \Theta^*\right\|_{\text{F}} = \kappa\zeta\left\|\widehat{\Delta}\right\|_{\text{F}}9\sqrt{r}\frac{\lambda}{\kappa} = \zeta\lambda\left\|\widehat{\Delta}\right\|_{\text{F}}9\sqrt{r},$$

and for the right-handed side, based on Eqn. (7) we have that

$$\langle\nabla\hat{L}_\tau(\widetilde{\Theta}_\zeta) - \nabla\hat{L}_\tau(\Theta^*), \widetilde{\Theta}_\zeta - \Theta^*\rangle \leq \zeta\langle\nabla\hat{L}_\tau(\widehat{\Theta}) - \nabla\hat{L}_\tau(\Theta^*), \widehat{\Theta} - \Theta^*\rangle$$

$$\leq \eta\frac{3\lambda}{2}\left\|\widehat{\Delta}\right\|_{\text{nuc}} \leq \zeta 6\sqrt{2}\lambda\sqrt{r}\left\|\widehat{\Delta}\right\|_{\text{F}}$$

Consequently, we have $9 \leq 6\sqrt{2}$ that contradicts the fact, which means that

$$\left\|\widehat{\Delta}\right\|_{\text{F}} \leq 9\sqrt{r} \cdot \frac{\lambda}{\kappa}.$$

$\square$

Next, we will show the event $\mathcal{E}(s, l, \kappa)$ and the event $\lambda \geq 2\left\|\nabla\hat{L}_\tau(\Theta^*)\right\|_{\text{op}}$ hold with high probability individually. Specifically, we will first give an upper bound of $\left\|\nabla\hat{L}_\tau(\Theta^*)\right\|_{\text{op}}$ in Theorem A.5 and then present the event $\mathcal{E}(s, l, \kappa)$ holds with high probability in Theorem A.6.

**Theorem A.5.** *By taking $\tau = \left(\frac{n}{5d-\ln(\epsilon)}\right)^{\frac{1}{1+\delta}}c^{\frac{1}{1+\delta}}$, then with probability at least $1 - \epsilon$, it holds that*

$$\left\|\nabla\hat{L}_\tau(\Theta^*)\right\|_{op} \leq (10 + 11\sqrt{2})\sigma\left(\frac{n}{5d - \ln(\epsilon)}\right)^{\frac{\delta}{1+\delta}}c^{\frac{1}{1+\delta}}.$$

*Proof.* Define the zero-mean random matrix $\Gamma = \nabla\hat{L}_\tau(\Theta^*) - \mathbb{E}\nabla\hat{L}_\tau(\Theta^*)$, then we have that

$$\left\|\nabla\hat{L}_\tau(\Theta^*)\right\|_{op} = \left\|\nabla\hat{L}_\tau(\Theta^*) - \mathbb{E}\nabla\hat{L}_\tau(\Theta^*) + \mathbb{E}\nabla\hat{L}_\tau(\Theta^*)\right\|_{op} \leq \|\Gamma\|_{op} + \left\|\mathbb{E}\nabla\hat{L}_\tau(\Theta^*)\right\|_{op}.$$

Therefore, we could control these two terms separately. Denote $S^{d-1} = \{u \in \mathbb{R}^d : \|u\|_2 = 1\}$. For the second term, we have that

$$\nabla\hat{L}_\tau(\Theta^*) = -\frac{1}{n}\sum_{i=1}^n l'_\tau(y_i - \langle X_i, \Theta^*\rangle)X_i = -\frac{1}{n}\sum_{i=1}^n l'_\tau(\eta_i)X_i.$$

Therefore, we can deduce that

$$\left\|\mathbb{E}\nabla\hat{L}_\tau(\Theta^*)\right\|_{op} = \sup_{u \in S^{d_1-1}, v \in S^{d_2-1}}\frac{1}{n}\sum_{i=1}^n\mathbb{E}\left(l'_\tau(\eta_i)u^\top X_i v\right)$$

$$= \sup_{u \in S^{d_1-1}, v \in S^{d_2-1}}\frac{1}{n}\sum_{i=1}^n\mathbb{E}\left(\mathbb{E}\left(l'_\tau(\eta_i)u^\top X_i v|\mathcal{F}_i\right)\right)$$

$$= \sup_{u \in S^{d_1-1}, v \in S^{d_2-1}}\frac{1}{n}\sum_{i=1}^n\mathbb{E}\left(u^\top X_i v \cdot \mathbb{E}\left(l'_\tau(\eta_i)|\mathcal{F}_i\right)\right)$$

By the expression of $l'_\tau(\cdot)$, we can deduce that

$$\left| \mathbb{E}\left( l'_\tau(\eta_i) | \mathcal{F}_i \right) \right| = \left| \mathbb{E}\left( l'_\tau(\eta_i) - \eta_i | \mathcal{F}_i \right) \right| \leq \mathbb{E}\left( \frac{|\eta_i|^{1+\delta}}{\tau^\delta} \middle| \mathcal{F}_i \right) \leq \frac{c}{\tau^\delta}$$

And since $u^\top X_i v$ is sub-Gaussian with the parameter $\sigma^2$, we have $\mathbb{E}(|u^\top X_i v|) \leq \sqrt{2\sigma^2}$. Conclusively, it holds that

$$\left\| \mathbb{E}\nabla\hat{L}_\tau(\Theta^*) \right\|_{op} \leq \frac{\sqrt{2}}{\tau^\delta} c \cdot \sigma. \tag{10}$$

To bound the operator norm of $\Gamma$, we use the regular covering technique: Let $\mathcal{N}_{\frac{1}{4}}^d$ be the $1/4$ covering of $S^{d-1}$, then we claim that

$$\|\Gamma\|_{op} \leq \frac{5}{2} \max_{u \in \mathcal{N}_{\frac{1}{4}}^{d_1}, v \in \mathcal{N}_{\frac{1}{4}}^{d_2}} u^\top \Gamma v. \tag{11}$$

To prove this result, for any $u \in S^{d_1-1}, v \in S^{d_2-1}$, we denote $S(u) \in \mathbb{R}^{d_1}$ ($S(v) \in \mathbb{R}^{d_2}$) as the nearest neighbor of $u$ ($v$) in $N_{\frac{1}{4}}^{d_1}$ ($\mathcal{N}_{\frac{1}{4}}^{d_2}$) such that $\|u - S(u)\|_2, \|v - S(v)\|_2 \leq \frac{1}{4}$. We take $u, v$ such that $u^\top \Gamma v = \|\Gamma\|_{op}$. Therefore, it holds that

$$\|\Gamma\|_{op} = u^\top \Gamma v = S(u)^\top \Gamma S(v) + (u - S(u))^\top \Gamma v + u^\top \Gamma(v - S(v)) + (u - S(u))^\top \Gamma(v - S(v))$$

$$\leq \max_{u \in \mathcal{N}_{\frac{1}{4}}^{d_1}, v \in \mathcal{N}_{\frac{1}{4}}^{d_2}} u^\top \Gamma v + \frac{1}{4} \|\Gamma\|_{op} + \frac{1}{4} \|\Gamma\|_{op} + \frac{1}{16} \|\Gamma\|_{op} \leq \max_{u \in \mathcal{N}_{\frac{1}{4}}^{d_1}, v \in \mathcal{N}_{\frac{1}{4}}^{d_2}} u^\top \Gamma v + \frac{3}{5} \|\Gamma\|_{op},$$

which leads to Eqn. (11). And then it holds that

$$\|\Gamma\|_{op} \leq \frac{5}{2} \max_{u \in \mathcal{N}_{\frac{1}{4}}^{d_1}, v \in \mathcal{N}_{\frac{1}{4}}^{d_2}} \frac{1}{n} \sum_{i=1}^n \left[ \mathbb{E}\left( l'_\tau(\eta_i) u^\top X_i v \right) - l'_\tau(\eta_i) u^\top X_i v \right].$$

To bound the right-hand side term, we aim to use a union bound of probability with Corollary A.2. Since $u^\top X_i v$ is sub-Gaussian with parameter $\sigma$ for arbitrary $u \in \mathcal{N}_{\frac{1}{4}}^{d_1}, v \in \mathcal{N}_{\frac{1}{4}}^{d_2}$, then we have that for $k = 2, 3, \ldots$

$$\mathbb{E}|u^\top X_i v|^k = \int_0^\infty \mathbb{P}\left( |u^\top X_i v|^k > t \right) dt \leq 2 \int_0^\infty \exp\left( -\frac{t^2}{2k\sigma^2} \right) dt \leq \frac{1}{2} \cdot k! \cdot (\sqrt{2}\sigma)^k.$$

The above results along with the fact that $|l'_\tau(\cdot)| \leq \tau$ can lead to the following inequality for $k = 2, 3, \ldots$:

$$\mathbb{E}\left| \sum_{i=1}^n l'_\tau(\eta_i) u^\top X_i v \right|^k \leq n \cdot \tau^{k-1-\delta} \mathbb{E}\left( |l'_\tau(\eta_i)|^{1+\delta} |u^\top X_i v|^k \right) \leq \frac{1}{2} \cdot k! \cdot \left( \sqrt{2}\sigma\tau \right)^{k-2} \cdot (2n\sigma^2\tau^{1-\delta}c).$$

Based on Corollary A.2, it holds that

$$\mathbb{P}\left( u^\top \Gamma v \geq 4\sqrt{x\sigma^2\tau^{1-\delta}c} \frac{1}{\sqrt{n}} + 4\sqrt{2}\sigma\tau \frac{x}{n} \right) \leq e^{-x}.$$

By taking the union bound on all $u \in \mathcal{N}_{\frac{1}{4}}^{d_1}, v \in \mathcal{N}_{\frac{1}{4}}^{d_2}$ and using the fact that $9^{d_1+d_2} \leq e^{5d}$, it holds that

$$\mathbb{P}\left( \|\Gamma\|_{op} \geq \frac{5}{2} \max_{u \in \mathcal{N}_{\frac{1}{4}}^{d_1}, v \in \mathcal{N}_{\frac{1}{4}}^{d_2}} u^\top \Gamma v \geq 10\sigma\sqrt{c}\sqrt{\frac{5d - \ln(\epsilon)}{n}} \tau^{\frac{1-\delta}{2}} + 10\sqrt{2}\sigma\tau \frac{5d - \ln(\epsilon)}{n} \right) \leq \epsilon. \tag{12}$$

Combining the results in Eqn. (10) and Eqn. (12), we have that

$$\mathbb{P}\left( \left\| \nabla\hat{L}_\tau(\Theta^*) \right\|_{op} \geq 10\sigma\sqrt{c}\sqrt{\frac{5d - \ln(\epsilon)}{n}} \tau^{\frac{1-\delta}{2}} + 10\sqrt{2}\sigma\tau \cdot \frac{5d - \ln(\epsilon)}{n} + \frac{\sqrt{2}}{\tau^\delta} c \cdot \sigma \right) \leq \epsilon.$$

By taking $\tau = \left(\frac{n}{5d - \ln(\epsilon)}\right)^{\frac{1}{1+\delta}} \cdot c^{\frac{1}{1+\delta}}$, we have that

$$\mathbb{P}\left(\left\|\nabla \hat{L}_\tau(\Theta^*)\right\|_{\mathrm{op}} \leq (10 + 11\sqrt{2})c^{\frac{1}{1+\delta}} \cdot \left(\frac{5d - \ln(\epsilon)}{n}\right)^{\frac{\delta}{1+\delta}}\right) \geq 1 - \epsilon.$$

$\square$

**Theorem A.6.** *For any $s, l > 0$, if we take $\tau$ and $n$ such that*

$$\tau \geq \max\left\{32\sigma^2 s\sqrt{\frac{1}{c_l}}, \left(\frac{64\sigma^2 c}{c_l}\right)^{\frac{1}{1+\delta}}\right\}$$

$$n \geq \max\left\{8\ln(9)(d_1 + d_2), \left(225\sigma\sqrt{\ln(9)(d_1 + d_2)}\frac{\tau l}{sc_l}\right)^2, \left(\frac{48\sigma^2}{c_l}\sqrt{-2\ln(\epsilon)}\right)^2, -\frac{\tau^2}{c_l s^2}\ln(\epsilon)\right\}.$$

*Then with probability at least $1 - \epsilon$, the local restricted strong convexity $\mathcal{E}(s, l, \kappa)$ holds with $\kappa = \frac{c_l}{4}$.*

*Proof.* Given the values of $s, l > 0$, for the sake of simplicity we denote the event $\Phi$ as $\Phi = \mathcal{M}(\Theta^*, s, l) = \{\Theta \in \mathbb{R}^{d_1 \times d_2} : \|\Theta - \Theta^*\|_F \leq s, \|\Theta - \Theta^*\|_{\mathrm{nuc}} \leq l\|\Theta - \Theta^*\|_F\}$. Since the Huber loss is convex and differentiable, we have

$$D(\Theta) := \langle \nabla \hat{L}_\tau(\Theta) - \nabla \hat{L}_\tau(\Theta^*), \Theta - \Theta^*\rangle$$

$$= \frac{1}{n}\sum_{i=1}^{n}(l'_\tau(y_i - \langle X_i, \Theta^*\rangle) - l'_\tau(y_i - \langle X_i, \Theta\rangle)) \cdot \langle X_i, \Theta - \Theta^*\rangle$$

$$\geq \frac{1}{n}\sum_{i=1}^{n}(l'_\tau(y_i - \langle X_i, \Theta^*\rangle) - l'_\tau(y_i - \langle X_i, \Theta\rangle)) \cdot \langle X_i, \Theta - \Theta^*\rangle \cdot \mathbb{1}_{\Xi_i(\Theta)},$$

where the last inequality holds since Huber loss is convex, and $\Xi_i(\Theta)$ is defined as

$$\Xi_i(\Theta) = \left\{|\eta_i| \leq \frac{\tau}{2}\right\} \cap \left\{|\langle X_i, \Theta - \Theta^*\rangle| \leq \frac{\tau}{2s}\|\Theta - \Theta^*\|_F\right\}.$$

Note whenever $\Theta \in \Phi$ and $\Xi_i(\Theta)$ hold we have that

$$|y_i - \langle X_i, \Theta\rangle| \leq |y_i - \langle X_i, \Theta^*\rangle| + \frac{\tau}{2s} \cdot \|\Theta - \Theta^*\|_F \leq \tau.$$

Since we have $l''_\tau(u) = 1$ with $|u| \leq \tau$, it holds that

$$D(\Theta) \geq \frac{1}{n}\sum_{i=1}^{n}\langle X_i, \Theta - \Theta^*\rangle^2 \cdot \mathbb{1}_{\Xi_i(\Theta)}.$$

Furthermore, we define the function $\phi_R(x)$ with some $R > 0$ as

$$\phi_R(x) = \begin{cases} x^2, & \text{if } |x| \leq \frac{R}{2}; \\ (x - R)^2, & \text{if } \frac{R}{2} < x \leq R; \\ (x + R)^2, & \text{if } -R \leq x < -\frac{R}{2}; \\ 0, & \text{otherwise.} \end{cases}$$

And we know $\phi_r(\cdot)$ is $R$-Lipschitz continuous with the properties that

$$\phi_{\alpha R}(\alpha x) = \alpha^2 \phi_R(x) \ \forall \alpha > 0, \ \text{and} \ x^2 \cdot \mathbb{1}_{|x| \leq R/2} \leq \phi_R(x) \leq x^2 \cdot \mathbb{1}_{|x| \leq R}.$$

Then we can deduce that

$$
\frac{D(\Theta)}{\|\Theta - \Theta^*\|_{\mathrm{F}}^2} \geq \frac{1}{n} \sum_{i=1}^{n} \left( \frac{\langle X_i, \Theta - \Theta^* \rangle}{\|\Theta - \Theta^*\|_{\mathrm{F}}} \right)^2 \cdot \mathbb{1}_{\Xi_i(\Theta)} \geq \frac{1}{n} \sum_{i=1}^{n} \phi_{\frac{\tau}{2s}} \left( \frac{\langle X_i, \Theta - \Theta^* \rangle}{\|\Theta - \Theta^*\|_{\mathrm{F}}} \right) \cdot \mathbb{1}_{\{|\eta_i| \leq \frac{\tau}{2}\}}
$$

$$
:= \frac{1}{n} \sum_{i=1}^{n} \beta_{\tau,s}(X_i, \Theta, \eta_i)
$$

$$
= \frac{1}{n} \sum_{i=1}^{n} \mathbb{E}\left(\beta_{\tau,s}(X_i, \Theta, \eta_i)\right) + \frac{1}{n} \sum_{i=1}^{n} \beta_{\tau,s}(X_i, \Theta, \eta_i) - \frac{1}{n} \sum_{i=1}^{n} \mathbb{E}\left(\beta_{\tau,s}(X_i, \Theta, \eta_i)\right)
$$

$$
\geq \frac{1}{n} \sum_{i=1}^{n} \mathbb{E}\left(\beta_{\tau,s}(X_i, \Theta, \eta_i)\right) - \sup_{\Theta \in \Phi} \left| \frac{1}{n} \sum_{i=1}^{n} \beta_{\tau,s}(X_i, \Theta, \eta_i) - \frac{1}{n} \sum_{i=1}^{n} \mathbb{E}\left(\beta_{\tau,s}(X_i, \Theta, \eta_i)\right) \right|
$$

$$
:= A_1 - A_2.
$$

For simplicity we write $\Delta = \Theta - \Theta^*$ as a function of $\Theta$. To lower bound the first term $A_1$, we have that for any $i \in [n]$,

$$
\mathbb{E}\left(\beta_{\tau,s}(X_i, \Theta, \eta_i)\right) \geq \mathbb{E}\left[ \left( \frac{\langle X_i, \Delta \rangle}{\|\Delta\|_{\mathrm{F}}} \right)^2 \cdot \mathbb{1}_{\{|\langle X_i, \Delta \rangle| \leq \frac{\tau}{4s}\|\Delta\|_{\mathrm{F}}\}} \cdot \mathbb{1}_{\{|\eta_i| \leq \frac{\tau}{2}\}} \right]
$$

$$
\geq \mathbb{E}\left( \frac{\langle X_i, \Delta \rangle}{\|\Delta\|_{\mathrm{F}}} \right)^2 - \mathbb{E}\left[ \left( \frac{\langle X_i, \Delta \rangle}{\|\Delta\|_{\mathrm{F}}} \right)^2 \cdot \mathbb{1}_{\{|\langle X_i, \Delta \rangle| > \frac{\tau}{4s}\|\Delta\|_{\mathrm{F}}\}} \right] - \mathbb{E}\left[ \left( \frac{\langle X_i, \Delta \rangle}{\|\Delta\|_{\mathrm{F}}} \right)^2 \cdot \mathbb{1}_{\{|\eta_i| > \frac{\tau}{2}\}} \right]
$$

$$
:= A_{11} - A_{12} - A_{13}.
$$

Based on Assumption 3.1, we have $A_{11} \geq c_l$. Furthermore, it holds that

$$
A_{12} \leq \sqrt{\mathbb{E}\left( \frac{\langle X_i, \Delta \rangle}{\|\Delta\|_{\mathrm{F}}} \right)^4} \cdot \sqrt{\mathbb{E}\left( \frac{\langle X_i, \Delta \rangle}{\|\Delta\|_{\mathrm{F}}} \right)^4 \Big/ \left( \frac{\tau}{4s} \right)^4} \leq 256\sigma^4 \cdot \frac{s^2}{\tau^2}
$$

$$
A_{13} \leq \left( \frac{2}{\tau} \right)^{1+\delta} \mathbb{E}\left( \frac{\langle X_i, \Delta \rangle}{\|\Delta\|_{\mathrm{F}}} \right)^2 \cdot \mathbb{E}|\eta_i|^{1+\delta} \leq \frac{16}{\tau^{1+\delta}} \sigma^2 \cdot c.
$$

By choosing that $\tau \geq \max\left\{ 32\sigma^2 s \sqrt{\frac{1}{c_l}}, \left( \frac{64\sigma^2 c}{c_l} \right)^{\frac{1}{1+\delta}} \right\}$, it holds that $A_{12} \leq \frac{c_l}{4}$ and $A_{13} \leq \frac{c_l}{4}$, which indicates that

$$
\mathbb{E}\left(\beta_{\tau,s}(X_i, \Theta, \eta_i)\right) \geq \frac{c_l}{2}, \quad \forall i \in [n],
$$

which implies that

$$
A_1 \geq \frac{c_l}{2} \tag{13}
$$

Afterward, we'd like to upper-bound the term $A_{12}$. Since we have that $\forall i \in [n]$

$$
0 \leq \beta_{\tau,s}(X_i, \Theta, \eta_i) \leq \frac{\tau^2}{16s^2}, \quad \mathbb{E}\left(\beta_{\tau,s}(X_i, \Theta, \eta_i)\right)^2 \leq \mathbb{E}\left( \frac{\langle X_i, \Delta \rangle}{\|\Delta\|_{\mathrm{F}}} \right)^4 \leq 16\sigma^4.
$$

Then based on the Bousquet's inequality [Bousquet, 2002], with probability at least $1 - \epsilon$ it holds that

$$
A_2 \leq \mathbb{E}A_2 + \sqrt{\mathbb{E}A_2} \cdot \frac{\tau}{2s} \sqrt{\frac{-\ln(\epsilon)}{n}} + 4\sigma^2 \sqrt{\frac{-2\ln(\epsilon)}{n}} + \frac{\tau^2}{16s^2} \frac{-\ln(\epsilon)}{3n}
$$

$$
\leq 2\mathbb{E}A_2 + 4\sigma^2 \sqrt{\frac{-2\ln(\epsilon)}{n}} + + \frac{\tau^2}{16s^2} \frac{-4\ln(\epsilon)}{3n}.
$$

To bound the first term $\mathbb{E}A_2$, we use the regular Rademacher symmetrization argument by defining a series of iid Rademacher

random variables $\{e_i\}$ with $\tilde{X}_i, \tilde{\eta}_i$ that are iid with $X_i, \eta_i$:

$$\mathbb{E}A_2 = \mathbb{E}\left[\sup_{\Theta \in \Phi}\left|\frac{1}{n}\sum_{i=1}^{n}\beta_{\tau,s}(X_i, \Theta, \eta_i) - \frac{1}{n}\sum_{i=1}^{n}\mathbb{E}\left(\beta_{\tau,s}(X_i, \Theta, \eta_i)\right)\right|\right]$$

$$\leq \mathbb{E}\left[\sup_{\Theta \in \Phi}\left|\left(\frac{1}{n}\sum_{i=1}^{n}\beta_{\tau,s}(X_i, \Theta, \eta_i) - \frac{1}{n}\sum_{i=1}^{n}\mathbb{E}\left(\beta_{\tau,s}(\tilde{X}_i, \Theta, \tilde{\eta}_i)\right)\right)e_i\right|\right]$$

$$\leq 2\mathbb{E}\left[\sup_{\Theta \in \Phi}\frac{1}{n}\sum_{i=1}^{n}\beta_{\tau,s}(X_i, \Theta, \eta_i)e_i\right].$$

Denote the event $c(l) := \left\{\Theta \in \mathbb{R}^{d_1 \times d_2} : \|\Theta - \Theta^*\|_{\text{nuc}} \leq l\|\Theta - \Theta^*\|_{\text{F}}\right\}$. Recall that we define as:

$$\beta_{\tau,s}(X_i, \Theta, \eta_i) = \phi_{\frac{\tau}{2s}}\left(\frac{\langle X_i, \Theta - \Theta^*\rangle}{\|\Theta - \Theta^*\|_{\text{F}}}\right)\cdot \mathbb{1}_{\{|\eta_i|\leq\frac{\tau}{2}\}} = \phi_{\frac{\tau}{2s}}\left(\frac{\langle X_i, \Theta - \Theta^*\rangle}{\|\Theta - \Theta^*\|_{\text{F}}}\cdot \mathbb{1}_{\{|\eta_i|\leq\frac{\tau}{2}\}}\right).$$

Define $c(t) = \frac{2s}{\tau}\phi_{\frac{\tau}{2s}}(t)$ and it is easy to show that $c(\cdot)$ is a 1-Lipschitz function. By using the Talagrand's concentration inequality [Wainwright, 2019], it holds that

$$\mathbb{E}A_2 \leq \frac{\tau}{s}\cdot\mathbb{E}\left[\sup_{\Theta \in c(l)}\frac{1}{n}\sum_{i=1}^{n}e_i\cdot\frac{2s}{\tau}\cdot\phi_{\frac{\tau}{2s}}\left(\frac{\langle X_i, \Theta - \Theta^*\rangle}{\|\Theta - \Theta^*\|_{\text{F}}}\cdot\mathbb{1}_{\{|\eta_i|\leq\frac{\tau}{2}\}}\right)\right]$$

$$\frac{\tau}{s}\cdot\mathbb{E}\left[\sup_{\Theta \in c(l)}\frac{1}{n}\sum_{i=1}^{n}e_i\cdot\frac{2s}{\tau}\cdot\frac{\langle X_i, \Theta - \Theta^*\rangle}{\|\Theta - \Theta^*\|_{\text{F}}}\cdot\mathbb{1}_{\{|\eta_i|\leq\frac{\tau}{2}\}}\right]$$

$$\leq \frac{\tau}{s}\cdot\mathbb{E}\left[\sup_{\Theta \in c(l)}\frac{1}{n}\left\|\sum_{i=1}^{n}e_iX_i\cdot\mathbb{1}_{\{|\eta_i|\leq\frac{\tau}{2}\}}\right\|_{\text{op}}\cdot\left\|\frac{\Theta - \Theta^*}{\|\Theta - \Theta^*\|_{\text{F}}}\right\|_{\text{nuc}}\right]$$

$$\leq \frac{\tau l}{sn}\cdot\mathbb{E}\left\|\sum_{i=1}^{n}e_iX_i\cdot\mathbb{1}_{\{|\eta_i|\leq\frac{\tau}{2}\}}\right\|_{\text{op}}.$$

By using the same technique in the proof of Theorem A.5, we can bound the operator norm by using the covering argument. Denote $\mathcal{N}_{\frac{1}{4}}^{d}$ be the $1/4$ covering of $S^{d-1}$, then it holds that

$$\mathbb{E}\left\|\sum_{i=1}^{n}e_iX_i\cdot\mathbb{1}_{\{|\eta_i|\leq\frac{\tau}{2}\}}\right\|_{\text{op}} \leq \frac{5}{2}\cdot\mathbb{E}\left[\max_{u\in\mathcal{N}_{\frac{1}{4}}^{d_1}, v\in\mathcal{N}_{\frac{1}{4}}^{d_2}}\sum_{i=1}^{n}e_iu^{\top}X_iv\cdot\mathbb{1}_{\{|\eta_i|\leq\frac{\tau}{2}\}}\right].$$

Note for any pair of $u\in\mathcal{N}_{\frac{1}{4}}^{d_1}, v\in\mathcal{N}_{\frac{1}{4}}^{d_2}$, we have that

$$\mathbb{E}\left(\sum_{i=1}^{n}e_iu^{\top}X_iv\cdot\mathbb{1}_{\{|\eta_i|\leq\frac{\tau}{2}\}}\right) = 0$$

$$\mathbb{E}\left(\sum_{i=1}^{n}|e_i|^k|u^{\top}X_iv|^k\cdot\mathbb{1}_{\{|\eta_i|\leq\frac{\tau}{2}\}}\right) \leq \mathbb{E}|u^{\top}X_iv|^k \leq \frac{1}{2}\cdot k!\cdot(\sqrt{2}\sigma)^{k-2}\cdot 2\sigma^2, \quad k = 2, 3, \ldots.$$

We can write the moment generating function $M(\lambda)$ of the random variable $\sum_{i=1}^{n}e_i\cdot u^{\top}X_iv\cdot\mathbb{1}_{\{|\eta_i|\leq\frac{\tau}{2}\}}$ as:

$$M(\lambda) = \mathbb{E}\left[\exp\left(\lambda\sum_{i=1}^{n}e_i\cdot u^{\top}X_iv\cdot\mathbb{1}_{\{|\eta_i|\leq\frac{\tau}{2}\}}\right)\right] = \prod_{i=1}^{n}\mathbb{E}\left[\exp\left(\lambda e_i\cdot u^{\top}X_iv\cdot\mathbb{1}_{\{|\eta_i|\leq\frac{\tau}{2}\}}\right)\right]$$

$$\leq \prod_{i=1}^{n}\left[1 + \frac{\lambda^2\cdot 2\sigma^2}{2} + \frac{\lambda^2\cdot 2\sigma^2}{2}\left(\sum_{k=3}^{\infty}(|\lambda|\sqrt{2}\sigma)^{k-2}\right)\right]$$

$$= \prod_{i=1}^{n}\left[1 + \frac{2\lambda^2\sigma^2}{2}\cdot\frac{1}{1 - \sqrt{2}\sigma|\lambda|}\right]$$

$$\leq \exp\left(n\lambda^2\sigma^2\frac{1}{1 - \sqrt{2}\sigma|\lambda|}\right), \quad |\lambda| \leq \frac{1}{\sqrt{2}\sigma}.$$

Therefore, it holds that for any $s_0 > 0$

$$\mathbb{E}\left[\max_{u \in \mathcal{N}_{\frac{1}{4}}^{d_1}, v \in \mathcal{N}_{\frac{1}{4}}^{d_2}} \sum_{i=1}^{n} e_i u^\top X_i v \cdot \mathbb{1}_{\{|\eta_i| \le \frac{\tau}{2}\}}\right] = \frac{1}{s_0} \mathbb{E}\left[\ln\left(\exp\left(s_0 \cdot \sum_{i=1}^{n} e_i u^\top X_i v \cdot \mathbb{1}_{\{|\eta_i| \le \frac{\tau}{2}\}}\right)\right)\right]$$

$$\le \frac{1}{s_0} \ln\left(\mathbb{E}\left[\max_{u \in \mathcal{N}_{\frac{1}{4}}^{d_1}, v \in \mathcal{N}_{\frac{1}{4}}^{d_2}} \exp\left(s_0 \cdot \sum_{i=1}^{n} e_i u^\top X_i v \cdot \mathbb{1}_{\{|\eta_i| \le \frac{\tau}{2}\}}\right)\right]\right)$$

$$\le \frac{1}{s_0} \ln\left(9^{d_1+d_2} \mathbb{E}\left[\exp\left(s_0 \cdot \sum_{i=1}^{n} e_i u^\top X_i v \cdot \mathbb{1}_{\{|\eta_i| \le \frac{\tau}{2}\}}\right)\right]\right)$$

$$= \frac{(d_1 + d_2)\ln(9) + ns_0^2\sigma^2 \cdot \frac{1}{1 - \sqrt{2}\sigma|s_0|}}{s_0}, \quad \forall |s_0| \le \frac{1}{\sqrt{2}\sigma}.$$

By taking $s_0 = \frac{\sqrt{(d_1+d_2)\ln(9)}}{\sigma \cdot \sqrt{n}}$, and conditioned on $n \ge 8\ln(9)(d_1 + d_2)$, we have that

$$\mathbb{E}\left[\max_{u \in \mathcal{N}_{\frac{1}{4}}^{d_1}, v \in \mathcal{N}_{\frac{1}{4}}^{d_2}} \sum_{i=1}^{n} e_i u^\top X_i v \cdot \mathbb{1}_{\{|\eta_i| \le \frac{\tau}{2}\}}\right] \le 3\sqrt{\ln(9)} \cdot \sqrt{n(d_1 + d_2)} \cdot \sigma.$$

And this fact implies that

$$\mathbb{E}A_2 \le \frac{15\tau\sigma l}{2s}\sqrt{\ln(9)}\sqrt{\frac{d_1 + d_2}{n}}.$$

Conclusively, with probability at least $1 - \epsilon$ we have that

$$A_2 \le \frac{15\tau\sigma l}{s}\sqrt{\ln(9)}\sqrt{\frac{d_1 + d_2}{n}} + 4\sigma^2\sqrt{\frac{-2\ln(\epsilon)}{n}} + + \frac{\tau^2}{16s^2}\frac{-4\ln(\epsilon)}{3n}.$$

Therefore, by ensuring that

$$n \ge \max\left\{8\ln(9)(d_1 + d_2), \left(225\sigma\sqrt{\ln(9)(d_1 + d_2)}\frac{\tau l}{sc_l}\right)^2, \left(\frac{48\sigma^2}{c_l}\sqrt{-2\ln(\epsilon)}\right)^2, -\frac{\tau^2}{c_l s^2}\ln(\epsilon)\right\},$$

we have

$$\mathbb{P}\left(A_2 \le \frac{c_l}{4}\right) \ge 1 - \epsilon. \tag{14}$$

Given the results shown in Eqn. (13) and Eqn. (14), we have that with probability at least $1 - \epsilon$, it holds that

$$\frac{\langle \nabla \hat{L}_\tau(\Theta) - \nabla \hat{L}_\tau(\Theta^*), \Theta - \Theta^* \rangle}{\|\Theta - \Theta^*\|_F^2} \ge \frac{c_l}{4}, \quad \forall \Theta \in \Phi.$$

$\square$

## A.2 PROOF OF THEOREM 4.1

Theorem 4.1 can be naturally proved based on the above Theorem A.4, Theorem A.5 and Theorem A.6. Here we assume $c_l$ and $\sigma$ are in constant scale in general, and for the LowHTR problem with $\sigma^2 \asymp c_l \asymp \frac{1}{d_1 d_2}$, our proof can be slightly modified as we discuss later.

By taking $\lambda \asymp \sigma\left(\frac{d - \ln(\epsilon)}{n}\right)^{\frac{\delta}{1+\delta}} c^{\frac{1}{1+\delta}}$, and $\tau \asymp \left(\frac{n}{d - \ln(\epsilon)}\right)^{\frac{1}{1+\delta}} c^{\frac{1}{1+\delta}}$, we can guarantee that $\lambda \ge 2\left\|\nabla\hat{L}_\tau(\Theta^*)\right\|_{\text{op}}$ with probability at least $1 - \epsilon$ from Theorem A.5. By choosing $l \asymp 4\sqrt{2r}$ and $s = \frac{\tau}{32\sigma^2}\sqrt{c_l}$, then the conditions in Theorem A.4 can be satisfied as long as $n \gtrsim (d - \ln(\epsilon))\sqrt{r\nu^3}$ where we denote $\nu = \frac{\sigma^2}{c_l}$. Furthermore, under the above setting, we know the local restricted strong convexity $\mathcal{E}(s, l, c_l/4)$ holds with probability at least $1 - \epsilon$ as long as the conditions in

Theorem A.6 hold. By reviewing the conditions of Theorem A.6, we know it suffices to have $n \gtrsim d, \nu^2, dr\nu^3$. Therefore, with probability at least $1 - 2\epsilon$, the final error bound in Theorem A.4 indicates that

$$\left\|\widehat{\Theta} - \Theta^*\right\|_{\mathrm{F}} \lesssim \frac{\sigma}{c_l} \left(\frac{d + \ln(1/\epsilon)}{n}\right)^{\frac{\delta}{1+\delta}} c^{\frac{1}{1+\delta}} \sqrt{r}.$$

$\square$

# B PROOF OF THEOREM 4.2

We now prove the regret bound given in Theorem 4.2: We have $\|\theta^*\| \leq S$ based on Section 3 and $\|\theta^*_{k+1:p}\| \leq S_\perp$ for some small $S_\perp$. In the beginning, we have the transformed buffer set $\mathcal{H}'_1$ of size $H := |\mathcal{H}'_1|$, and we write the pair information $(X, y)$ in $\mathcal{H}'_1$ as $\{(x_{s,1}, y_{s,1}), \ldots, (x_{s,H}, y_{s,H})\}$. And we denote $(x_{e,t}, y_{e,t})$ as the pair of pulled arm and corresponding stochastic payoff at round $t$. To abuse the notation, at round $t + 1$ we denote $\{(x_i, y_i)\}_{i=1}^{t+H}$ as the pairs of observations in the initial buffer set and obtained by the end of round $t$ in order.

At the beginning of the round $t + 1$, the current $M := M_t$ can be written as $M_t = \sum_{i=1}^{H} x_{s,i} x_{s-i}^\top \sum_{j=1}^{t} x_{e,j} x_{e,j}^\top + \Lambda$, where $\Lambda$ is a positive diagonal matrix with $\lambda$ occupying the first $k$ diagonal entries and $\lambda_\perp$ the next $p - k$ entries. According to Algorithm 2, we denote $X_t \in \mathbb{R}^{(t+H) \times p}$ where each row of $X_t$ is the feature vector of the pulled arm (in the history buffer set or not). Assume $t + H > p$, we denote its full SVD as $X_t = U_x \Sigma_x V_x^\top$ with $U_x \in \mathbb{R}^{(t+H) \times p}$ and $V_x \in \mathbb{R}^{p \times p}$. We also write $M_t = V_x(\Sigma^2 + \Lambda)V_x^\top \in \mathbb{R}^{p \times p}$. And we further denote

$$\begin{pmatrix} u_1^\top \\ u_2^\top \\ \vdots \\ u_p^\top \end{pmatrix} = M_t^{-\frac{1}{2}} X_t^\top = V_x(\Sigma_x^2 + \Lambda)^{-\frac{1}{2}} \cdot \Sigma_x U_x^\top \preceq V_x U_x^\top = \begin{pmatrix} V_{x,11} & \cdots & V_{x,1p} \\ \vdots & \ddots & \vdots \\ V_{x,p1} & \cdots & V_{x,pp} \end{pmatrix} \cdot \begin{pmatrix} U_{x,1}^\top \\ \vdots \\ U_{x,p}^\top \end{pmatrix} \in \mathbb{R}^{p \times (t+H)}.$$

We first show that for all $i \in [p]$,

$$\|u_i\|_2 \leq \|\sum_{j=1}^{p} V_{ij} U_j\|_2 = \sqrt{\sum_{j=1}^{p} V_{ij}^2 \|U_j\|_2^2} = 1$$

$$\|u_i\|_{1+\delta} \leq (t + H)^{\frac{1}{1+\delta} - \frac{1}{2}} \cdot \|u_i\|_2 \leq (t + H)^{\frac{1-\delta}{2(1+\delta)}},$$

where the last inequality is deduced from the Cauchy-Schwarz inequality. With the formulation of $\hat{\theta}_t$ in Algorithm 2 line 3, we have that

$$\begin{aligned}
\|\hat{\theta}_t - \theta^*\|_{M_t} &= \left\| M_t^{-\frac{1}{2}} \begin{pmatrix} u_1^\top \hat{y}_1 \\ \vdots \\ u_p^\top \hat{y}_p \end{pmatrix} - M_t^{-1} X_t^\top X_t \theta^* - M_t^{-1} \Lambda \theta^* \right\|_{M_t} \\
&\leq \left\| M_t^{-\frac{1}{2}} \begin{pmatrix} u_1^\top \hat{y}_1 \\ \vdots \\ u_p^\top \hat{y}_p \end{pmatrix} - M_t^{-\frac{1}{2}} \begin{pmatrix} u_1^\top \\ \vdots \\ u_p^\top \end{pmatrix} X_t \theta^* \right\|_{M_t} + \|\Lambda \theta^*\|_{M_t^{-1}} \\
&\leq \left\| \begin{pmatrix} u_1^\top (\hat{y}_1 - X_t \theta^*) \\ \vdots \\ u_p^\top (\hat{y}_p - X_t \theta^*) \end{pmatrix} \right\|_2 + \|\theta^*\|_\Lambda \\
&\leq \sqrt{\sum_{i=1}^{p} \left(u_i^\top (\hat{y}_i - X_t \theta^*)\right)^2} + \sqrt{\lambda_0} S + \sqrt{\lambda_\perp} S_\perp.
\end{aligned}$$

To present a bound on the first term, we divide it into two separate parts.

$$u_i^\top (\hat{y}_i - X_t \theta^*) = \sum_{j=1}^{t+H} u_{i,j}(\hat{y}_{i,j} - \mathbb{E}(y_j | \mathcal{F}_{j-1}))$$

$$= \sum_{j=1}^{t+H} u_{i,j} \left[ (\hat{y}_{i,j} - \mathbb{E}(\hat{y}_{i,j} | \mathcal{F}_{j-1})) - \mathbb{E}(y_j \mathbb{1}_{\{|u_{i,j} y_j| > b_t\}} | \mathcal{F}_{j-1}) \right]$$

$$\leq \left| \sum_{j=1}^{t+H} u_{i,j}(\hat{y}_{i,j} - \mathbb{E}(\hat{y}_{i,j} | \mathcal{F}_{j-1})) \right| + \left| \sum_{j=1}^{t+H} u_{i,j} \mathbb{E}(y_j \mathbb{1}_{\{|u_{i,j} y_j| > b_t\}} | \mathcal{F}_{j-1}) \right| := A_1 + A_2$$

For the first term $A_1$, based on Bernstein' inequality for martingales [Seldin et al., 2012], for any $i \in [p]$ it holds that with probability at least $1 - \frac{\epsilon}{p}$:

$$A_1 \leq 2b_t \ln \left( \frac{2p}{\epsilon} \right) + \left| \frac{1}{2b_t} \sum_{j=1}^{t+H} \mathbb{E} \left[ u_{i,j}^2 (\hat{y}_{i,j} - \mathbb{E}(\hat{y}_{i,j} | \mathcal{F}_{j-1}))^2 | \mathcal{F}_{j-1} \right] \right|$$

$$\leq 2b_t \ln \left( \frac{2p}{\epsilon} \right) + \frac{b_t}{2} \left| \sum_{j=1}^{t+H} \mathbb{E} \left[ \left( \frac{u_{i,j}(\hat{y}_{i,j} - \mathbb{E}(\hat{y}_{i,j} | \mathcal{F}_{j-1}))}{b_t} \right)^2 | \mathcal{F}_{j-1} \right] \right| := 2b_t \ln \left( \frac{2p}{\epsilon} \right) + \frac{b_t}{2} \left| \sum_{j=1}^{t+H} \mathbb{E} \left[ T | \mathcal{F}_{j-1} \right] \right|.$$

Since we know that $|T| \leq 1$ and hence $\mathbb{E}(T^2) \leq \mathbb{E}(|T|^{1+\delta})$, and we can then deduce that

$$A_1 \leq 2b_t \ln \left( \frac{2p}{\epsilon} \right) + \frac{b_t}{2} \cdot \frac{\sum_{j=1}^{t+H} |u_{i,j}|^{1+\delta} \cdot b}{b_t^{1+\delta}} \leq 2b_t \ln \left( \frac{2p}{\epsilon} \right) + \frac{b}{2b_t^\delta}(t+H)^{\frac{1-\delta}{2}}.$$

Therefore, we know that with probability at least $1 - \epsilon$ the following result holds for all $i \in [p]$ simultaneously:

$$\left| \sum_{j=1}^{t+H} u_{i,j}(\hat{y}_{i,j} - \mathbb{E}(\hat{y}_{i,j} | \mathcal{F}_{j-1})) \right| \leq 2b_t \ln \left( \frac{2p}{\epsilon} \right) + \frac{b}{2b_t^\delta}(t+H)^{\frac{1-\delta}{2}}.$$

For the term $A_2$, with the help of Holder's inequality, we have for all $i \in [p]$:

$$A_2 \leq \sum_{j=1}^{t+H} \mathbb{E} \left( |u_{i,j} y_j|^{1+\delta} \right)^{\frac{1}{1+\delta}} \cdot \mathbb{E} \left( \mathbb{1}_{|u_{i,j} y_j| > b_t} \right)^{\frac{\delta}{1+\delta}}$$

$$\leq \sum_{j=1}^{t+H} |u_{i,j}| \cdot b^{\frac{1}{1+\delta}} \cdot \mathbb{P} \left( |u_{i,j} y_j| > b_t \right)^{\frac{\delta}{1+\delta}}$$

$$\leq \sum_{j=1}^{t+H} |u_{i,j}| \cdot b^{\frac{1}{1+\delta}} \cdot \left( \frac{|u_{i,j}|^{1+\delta} b}{b_t^{1+\delta}} \right) \leq \frac{b}{b_t^\delta} \cdot (t+H)^{\frac{1-\delta}{2}}.$$

Therefore, by taking

$$b_t = \left( \frac{b}{\ln \left( \frac{2p}{\epsilon} \right)} \right)^{\frac{1}{1+\delta}} \cdot (t+H)^{\frac{1-\delta}{2+2\delta}},$$

we can deduce that with probability at least $1 - \epsilon$ the following result holds for all $i \in [p]$ simultaneously:

$$u_i^\top (\hat{y}_i - X_t \theta^*) \leq 4b^{\frac{1}{1+\delta}} \left( \ln \left( \frac{2p}{\delta} \right) \right)^{\frac{\delta}{1+\delta}} \cdot (t+H)^{\frac{1-\delta}{2+2\delta}}.$$

Therefore, with probability at least $1 - \epsilon$ it holds that

$$\|\hat{\theta}_t - \theta^*\|_{M_t} \leq 2\sqrt{p} \cdot b^{\frac{1}{1+\delta}} \left( \ln \left( \frac{2p}{\delta} \right) \right)^{\frac{\delta}{1+\delta}} \cdot (t+H)^{\frac{1-\delta}{2+2\delta}} := \beta_t(\epsilon).$$

Denote the optimal arm at time $t+1$ as $x^*_{e,t+1}$. Therefore, the instance regret at time $t+1$ can be bounded by

$$x^*_{e,t+1}{}^\top \theta^* - x^\top_{e,t+1}\theta^* = x^*_{e,t+1}{}^\top \theta^* - x^*_{e,t+1}{}^\top \hat\theta_t + x^*_{e,t+1}{}^\top \hat\theta_t - x^\top_{e,t+1}\hat\theta_t + x^\top_{e,t+1}\hat\theta_t - x^\top_{e,t+1}\theta^*$$
$$\le \beta_t(\epsilon)\|x^*_{e,t+1}\|_{M_t^{-1}} + x^\top_{e,t+1}\hat\theta_t + \beta_t(\epsilon)\|x_{e,t+1}\|_{M_t^{-1}} - x^\top_{e,t+1}\hat\theta_t - \|x^*_{e,t+1}\|_{M_t^{-1}} + \beta_t(\epsilon)\|x_{e,t+1}\|_{M_t^{-1}}$$
$$\le \min\{S^2, 2\beta_t(\epsilon)\|x_{e,t+1}\|_{M_t^{-1}}\}.$$

Therefore, with probability at least $1-\epsilon$, it holds that

$$\sum_{t=1}^{T} r_t = \sum_{t=1}^{T} \min\{S^2, 2\beta_t\left(\frac{\epsilon}{T}\right)\|x_{e,t+1}\|_{M_t^{-1}}\}$$

$$\le 2\beta_T\left(\frac{\epsilon}{T}\right)\sum_{t=1}^{T}\min\{\frac{S^2}{\beta_T\left(\frac{\epsilon}{T}\right)}, \|x_{e,t+1}\|_{M_t^{-1}}\} \le 2\beta_T\left(\frac{\epsilon}{T}\right)\cdot \sqrt{T}\cdot \sqrt{\sum_{t=1}^{T}\min\{\|x_{e,t+1}\|^2_{M_t^{-1}}, 1\}}$$

We denote $\tilde M_{T+1} = \sum_{t=1}^{T} x_{e,t}x^\top_{e,t} + \Lambda$, and by Lemma 9 of Dani et al. [2008], it holds that

$$\sqrt{\sum_{t=1}^{T}\min\{\|x_{e,t+1}\|^2_{M_t^{-1}}, 1\}} \le 2\ln\left(\frac{\det(\tilde M_{T+1})}{\det(\Lambda)}\right) \le 2k\cdot\ln\left(1+\frac{S^2}{k\lambda_0}T\right) + 2(p-k)\ln\left(1+\frac{S^2}{(p-k)\lambda_\perp}T\right)$$

$$\le 2k\cdot\ln\left(1+\frac{S^2}{k\lambda_0}T\right) + \frac{2S^2}{\lambda_\perp}T \le 4k\cdot\ln\left(1+\frac{S^2}{k\lambda_0}T\right),$$

by taking that $\lambda_\perp = \frac{S^2 T}{k\ln\left(1+\frac{S^2}{k\lambda_0}T\right)}$. Therefore, with probability at least $1-\epsilon$, it holds that

$$R(T) \le 2\sqrt{T}\cdot\sqrt{4k\cdot\ln\left(1+\frac{S^2}{k\lambda_0}T\right)}\cdot\left[2\sqrt{p}\cdot b^{\frac{1}{1+\delta}}\left(\ln\left(\frac{2p}{\delta}\right)\right)^{\frac{\delta}{1+\delta}}\cdot(T+H)^{\frac{1-\delta}{2+2\delta}} + \sqrt{\lambda_0}S + \sqrt{\lambda_\perp}S_\perp\right]$$

$$= \tilde O\left(\sqrt{kp}\cdot T^{\frac{1}{1+\delta}} + \sqrt{kT} + S_\perp T\right).$$

$\square$

## C  PROOF OF EQN. (6)

Our argument is adapted from the proof of Theorem 3 in Jun et al. [2019], and we will still present details here for completeness of our work. Furthermore, the proof of Theorem 4.4 in our work still relies on the same Lemma.

**Lemma C.1.** (Wedin's $\sin\Theta$ Theorem) *Let the SVDs of matrices $A$ and $\tilde A$ be defined as follows:*

$$\begin{pmatrix} U_1 & U_2 & U_3 \end{pmatrix}^\top A \begin{pmatrix} V_1 & V_2 \end{pmatrix} = \begin{pmatrix} \Sigma_1 & 0 \\ 0 & \Sigma_2 \\ 0 & 0 \end{pmatrix},$$

$$\begin{pmatrix} \tilde U_1 & \tilde U_2 & \tilde U_3 \end{pmatrix}^\top \tilde A \begin{pmatrix} \tilde V_1 & \tilde V_2 \end{pmatrix} = \begin{pmatrix} \tilde\Sigma_1 & 0 \\ 0 & \tilde\Sigma_2 \\ 0 & 0 \end{pmatrix}.$$

*Let $R = A\tilde V_1 - \tilde U_1\tilde\Sigma_1$ and $S = A^\top\tilde U_1 - \tilde V_1\tilde\Sigma_1$, and define $U_{1\perp} = [U_2\ U_3]$ and $V_{1\perp} = [V_2\ V_3]$. Then suppose there is a number $q > 0$ such that*

$$\min_{i,j}|\sigma_i(\tilde\Sigma_1) - \sigma_j(\Sigma_2)| \ge q, \quad \min_i \sigma_i(\tilde\Sigma_1) \ge q,$$

*Then it holds that*

$$\sqrt{\left\|U^\top_{1\perp}\tilde U_1\right\|^2_F + \left\|V^\top_{1\perp}\tilde V_1\right\|^2_F} \le \frac{\sqrt{\|R\|^2_F + \|S\|^2_F}}{q}.$$

Based on Lemma C.1, we define $A = \widehat{\Theta}, U_1 = \widehat{U}, \Sigma_1 = \widehat{D}, V_1 = \widehat{V}, \tilde{A} = \Theta^*, \tilde{U}_1 = U, \tilde{\Sigma}_1 = D, \tilde{V}_1 = V, q = D_{rr}$. Therefore, according to Lemma C.1, we have that $R = (\widehat{\Theta} - \Theta^*)\widehat{V}$ and $S = -(\widehat{\Theta} - \Theta^*)^\top U$, and then it holds that

$$\sqrt{2 \left\|\widehat{U}_\perp^\top U\right\|_{\mathrm{F}} \left\|\widehat{V}_\perp^\top V\right\|_{\mathrm{F}}} \le \sqrt{\left\|\widehat{U}_\perp^\top U\right\|_{\mathrm{F}}^2 + \left\|\widehat{V}_\perp^\top V\right\|_{\mathrm{F}}^2} \le \frac{\sqrt{\|R\|_{\mathrm{F}}^2 + \|S\|_{\mathrm{F}}^2}}{D_{rr}} \le \frac{\sqrt{2} \cdot \left\|\widehat{\Theta} - \Theta^*\right\|_{\mathrm{F}}}{D_{rr}}.$$

And then by using the bound on $\left\|\widehat{\Theta} - \Theta^*\right\|_{\mathrm{F}}$ we can deduce that

$$\|\theta_{k+1:p}^*\|_2 = \left\|\widehat{U}_\perp^\top U D V^\top \widehat{V}_\perp\right\|_{\mathrm{F}} \le \left\|\widehat{U}_\perp^\top U\right\|_{\mathrm{F}} \left\|\widehat{V}_\perp^\top V\right\|_{\mathrm{F}} \cdot \|D\|_{\mathrm{op}} \lesssim \frac{r\sigma^2 c^{\frac{2}{1+\delta}}}{c_l^2 D_{rr}^2} \left(\frac{d + \ln(1/\epsilon)}{|\mathcal{H}_2|}\right)^{\frac{2\delta}{1+\delta}}.$$

$\square$

# D PROOF OF THEOREM 4.3

We now prove Theorem 4.3 in this section. We first bring up the result shown in Eqn. (3) again: under Assumption 3.1, if we estimate $\Theta^*$ based on the exploration set $\mathcal{H}_2$ of size $H$, then our estimator $\widehat{\Theta}$ satisfies the following property:

$$\|\theta_{k+1:p}^*\|_2 \lesssim \frac{rd^2 c^{\frac{2}{1+\delta}}}{D_{rr}^2} \left(\frac{d + \ln(1/\epsilon)}{H}\right)^{\frac{2\delta}{1+\delta}},$$

under $\sigma^2 \asymp c_l \asymp 1/(d_1 d_2)$ with probability at least $1 - \epsilon$. Our Algorithm 1 first randomly samples arms for the first $T_1$ rounds, and then for the rest of the time horizon it utilizes a doubling-trick-based idea. Based on line 3 of Algorithm 1, when we have that

$$\left[\frac{d^{2+4\delta} r^{1+\delta}}{D_{rr}^{2+2\delta}} 2^{i(1+\delta)}\right]^{\frac{1}{1+3\delta}} \ge 2^i \implies i \le \left\lfloor \log_2 \left(\frac{d^{\frac{1+2\delta}{\delta}} r^{\frac{1+\delta}{2\delta}}}{D_{rr}^{\frac{1+\delta}{\delta}}}\right)\right\rfloor := L,$$

then in the first $L$ batches, we will run out of time to do random exploration. Since we have that

$$\frac{2d^{\frac{1+2\delta}{\delta}} r^{\frac{1+\delta}{2\delta}}}{D_{rr}^{\frac{1+\delta}{\delta}}} \ge \sum_{j=1}^{L} 2^j = 2^{L+1} - 2 \ge \frac{d^{\frac{1+2\delta}{\delta}} r^{\frac{1+\delta}{2\delta}}}{D_{rr}^{\frac{1+\delta}{\delta}}} - 2,$$

we know before the batch $L + 1$, we already repeat random sampling for $T_{\mathrm{init}}$ rounds, with

$$T_1 + \frac{d^{\frac{1+2\delta}{\delta}} r^{\frac{1+\delta}{2\delta}}}{D_{rr}^{\frac{1+\delta}{\delta}}} - 2 \le T_{\mathrm{init}} \le T_1 + \frac{2d^{\frac{1+2\delta}{\delta}} r^{\frac{1+\delta}{2\delta}}}{D_{rr}^{\frac{1+\delta}{\delta}}}.$$

For the sake of simplicity in our proof, we assume that our algorithm terminates exactly at the end of some batch, i.e. the $M$-th batch. And otherwise, our proof will be the same by using the index of the last batch. In other words, it holds that

$$\sum_{i=L+1}^{M} 2^i + T_{\mathrm{init}} = T \iff 2^{M+1} = T + 2^{L+1} - T_{\mathrm{init}}.$$

Therefore, if we set $\epsilon$ as $\epsilon/2^{i+1}$ in both $\beta_t$ of Algorithm 2 and $\lambda, \tau$ in the matrix estimation for the $i$-th batch, then based on Theorem 4.2, with probability at least $1 - \epsilon$ it holds that

$$R(T) = \widetilde{O}\left(T_{\mathrm{init}} + \sum_{i=L+1}^{M} \left[C\left(2^{\frac{1+\delta}{1+3\delta}}\right)^i + \sqrt{d^3 r}\left(2^{\frac{1}{1+\delta}}\right)^i + \sqrt{dr}2^i + 2^i \cdot \frac{d^{\frac{2+4\delta}{1+\delta}} r}{D_{rr}^2} \cdot \left(\frac{1}{T_{\mathrm{init}} + \sum_{j=L+1}^{i} C\left(2^{\frac{1+\delta}{1+3\delta}}\right)^j}\right)^{\frac{2\delta}{1+\delta}}\right]\right)$$

$$= \widetilde{O}\left(A_1 + \sum_{i=L+1}^{M} [A_{i,2} + A_{i,3} + A_{i,4} + A_{i,5}]\right),$$

with $C = \left(\frac{d^{2+4\delta}r^{1+\delta}}{D_{rr}^{2+2\delta}}\right)^{\frac{1}{1+3\delta}}$. For $A_1$, it naturally holds that $A_1 \lesssim T_{\text{init}}$. For $A_{i,2}$, we have that

$$\sum_{i=L+1}^{M} A_{i,2} \lesssim C \cdot \frac{1}{2^{\frac{1+\delta}{1+3\delta}} - 1} \cdot T^{\frac{1+\delta}{1+3\delta}}.$$

For $A_{i,3}$, we have that

$$\sum_{i=L+1}^{M} A_{i,3} \lesssim \sqrt{d^3 r} \frac{1}{2^{\frac{1}{1+\delta}} - 1} \cdot (T - T_{\text{init}})^{\frac{1}{1+\delta}} \lesssim \sqrt{d^3 r} \cdot T^{\frac{1}{1+\delta}}.$$

For $A_{i,3}$, it holds that

$$\sum_{i=L+1}^{M} A_{i,4} \lesssim \sqrt{dr}\sqrt{2^i} \lesssim \sqrt{dr} \cdot \frac{1}{\sqrt{2} - 1} \cdot (T - T_{\text{init}})^{\frac{1}{2}} \lesssim \sqrt{drT}.$$

And finally for $A_{i,5}$ we can show that

$$\sum_{i=L+1}^{M} A_{i,5} = \sum_{i=L+1}^{M} 2^i \cdot \frac{d^{\frac{2+4\delta}{1+\delta}} r}{D_{rr}^2} \cdot \left(\frac{1}{T_{\text{init}} + \sum_{j=L+1}^{i} C\left(2^{\frac{1+\delta}{1+3\delta}}\right)^j}\right)^{\frac{2\delta}{1+\delta}}$$

$$\lesssim \sum_{i=L+1}^{M} 2 \cdot C \cdot \left(\frac{\left(2^{\frac{1+\delta}{2\delta}}\right)^i}{\frac{T_1 - 2}{C} + \frac{d^{\frac{1+2\delta}{\delta}} r^{\frac{1+\delta}{2\delta}}}{D_{rr}^{\frac{1+\delta}{\delta}} C} + \sum_{j=L+1}^{i}\left(2^{\frac{1+\delta}{1+3\delta}}\right)^j}\right)^{\frac{2\delta}{1+\delta}}$$

$$\lesssim 2 \cdot C \cdot \sum_{L+1}^{M} \left[\frac{\left(2^{\frac{1+\delta}{1+3\delta}} - 1\right)^{\frac{2\delta}{1+\delta}}}{2^{\frac{(1+\delta)(2\delta)}{(1+3\delta)(1+\delta)}}} \cdot 2^{\left(\frac{1+\delta}{1+3\delta}\right)i}\right] \lesssim C \cdot T^{\frac{1+\delta}{1+3\delta}},$$

given that

$$T_1 \geq 2 - \frac{d^{\frac{1+2\delta}{\delta}} r^{\frac{1+\delta}{2\delta}}}{D_{rr}^{\frac{1+\delta}{\delta}}} + \left(\frac{d^{\frac{1+2\delta}{\delta}} r^{\frac{1+\delta}{2\delta}}}{D_{rr}^{\frac{1+\delta}{\delta}}}\right)^{\frac{1+\delta}{1+3\delta}} \cdot \frac{1}{2^{\frac{1+\delta}{1+3\delta}} - 1} \cdot C \geq 2 + \left(\frac{2}{\sqrt{2} - 1}\right) \cdot \frac{d^{\frac{1+2\delta}{\delta}} r^{\frac{1+\delta}{2\delta}}}{D_{rr}^{\frac{1+\delta}{\delta}}}.$$

Therefore, with the above condition on $T_1$ satisfied, the following result holds with probability at least $1 - \epsilon$

$$R(T) = \widetilde{O}\left(\frac{d^{\frac{2+4\delta}{1+3\delta}} r^{\frac{1+\delta}{1+3\delta}}}{D_{rr}^{\frac{2+2\delta}{1+3\delta}}} \cdot T^{\frac{1+\delta}{1+3\delta}} + d^{\frac{3}{2}} r^{\frac{1}{2}} T^{\frac{1}{1+\delta}}\right).$$

$\square$

# E PROOF OF THEOREM 4.4

The proof of Theorem 4.4 is adapted from that of Theorem 4.3 presented in the above Appendix D. According to Li [1998], it holds that

$$|\sigma_i(\widehat{\Theta}) - \sigma_i(\Theta^*)| \leq \left\|\widehat{\Theta} - \Theta^*\right\|_{\text{F}}, \quad \forall i \in [d].$$

Denote $H$ as the size of the exploration buffer set $\mathcal{H}_2$ at the end of the exploration phase for the $i-$th batch, then according to Theorem 4.1 we know that

$$\left\|\widehat{\Theta} - \Theta^*\right\|_{\text{F}} \leq C_1 \frac{\sigma\sqrt{r}}{c_l}\left(\frac{d + \ln\left(2^{i+1}/\epsilon\right)}{H}\right)^{\frac{\delta}{1+\delta}} \cdot c^{\frac{1}{1+\delta}} := E, \quad C_1 > 0, \tag{15}$$

with probability at least $1 - \epsilon/2^{i+1}$. We define the useful rank $\hat{r}$ as:

$$\hat{r} = \min\left\{ i \in [d+1] : \hat{D}_{ii} \leq C_1 \frac{\sigma\sqrt{i}}{c_l}\left(\frac{d + \ln(2^{i+1}/\epsilon)}{H}\right)^{\frac{\delta}{1+\delta}} \cdot c^{\frac{1}{1+\delta}} := R(i)\right\} - 1 \wedge 1,$$

We will first show that $\hat{D}_{(r+1)(r+1)} \leq R(r+1)$ and hence $\hat{r} \leq r$ holds if we have Eqn. (15). This is because that $\hat{D}_{(r+1)(r+1)} \leq E = R(r) < R(r+1)$. Furthermore, we will illustrate that all the subspaces we remove based on our estimated $\hat{r}$ are sufficiently minimal. Specifically, we know that

$$D_{(\hat{r}+1)(\hat{r}+1)} \leq \hat{D}_{(\hat{r}+1)(\hat{r}+1)} + |\hat{D}_{(\hat{r}+1)(\hat{r}+1)} - D_{(\hat{r}+1)(\hat{r}+1)}| \leq R(\hat{r}+1) + E \leq 2R(r+1).$$

To abuse the notation, we rewrite the SVD of $\widehat{\Theta}$ and $\Theta^*$ as

$$\widehat{\Theta} = \begin{pmatrix} \widehat{U} & \widehat{U}_r & \widehat{U}_\perp \end{pmatrix} \cdot \begin{pmatrix} \widehat{D}_{\hat{r}} & 0 & 0 \\ 0 & \widehat{D}_{r-\hat{r}} & 0 \\ 0 & 0 & \widehat{D}_0 \end{pmatrix} \cdot \begin{pmatrix} \widehat{V}^\top \\ \widehat{V}_r^\top \\ \widehat{V}_\perp^\top \end{pmatrix}$$

$$\Theta^* = \begin{pmatrix} \tilde{U} & \tilde{U}_r & \tilde{U}_\perp \end{pmatrix} \cdot \begin{pmatrix} \tilde{D}_{\hat{r}} & 0 & 0 \\ 0 & \tilde{D}_{r-\hat{r}} & 0 \\ 0 & 0 & 0 \end{pmatrix} \cdot \begin{pmatrix} \tilde{V}^\top \\ \tilde{V}_r^\top \\ \tilde{V}_\perp^\top \end{pmatrix}.$$

And by making sure that $H$ is sufficiently large such that $R(r+1) \leq D_{rr}/2$, we have that

$$\min|\sigma_i(D_{\hat{r}}) - \sigma_j(D_{r-\hat{r}})| \geq \frac{D_{rr}}{2}, \quad \min\sigma_i(D_{\hat{r}}) \geq D_{rr}.$$

In Lemma C.1, with $A = \widehat{\Theta}, U_1 = \widehat{U}, U_{1\perp} = [\widehat{U}_r, \widehat{U}_\perp], \Sigma_1 = \widehat{D}, V_1 = \widehat{V}, V_{1\perp} = [\widehat{V}_r, \widehat{V}_\perp], \tilde{A} = \Theta^*, \tilde{U}_1 = \tilde{U}, \tilde{\Sigma}_1 = D, \tilde{V}_1 = \tilde{V}, q = D_{rr}/2$, we can show that

$$\left\|\widehat{U}_{1\perp}^\top \tilde{U}\right\|_F \left\|\widehat{V}_{1\perp}^\top \tilde{V}\right\|_F \leq \frac{4\left\|\widehat{\Theta} - \Theta^*\right\|_F^2}{D_{rr}^2}.$$

After we do the same transformation in Algorithm 2, we know the effective dimension (denoted by $\hat{k}$) satisfies that $\hat{k} = d_1 d_2 - (d_1 - \hat{r})(d_2 - \hat{r}) \leq d_1 d_2 - (d_1 - r)(d_2 - r) = k$. And it holds that

$$\|\theta^*_{\hat{k}+1:p}\|_2 = \left\|U_{1\perp}^\top \begin{pmatrix} \tilde{U} & \tilde{U}_r \end{pmatrix} \cdot \begin{pmatrix} D_{\hat{r}} & 0 \\ 0 & D_{r-\hat{r}} \end{pmatrix} \cdot \begin{pmatrix} \tilde{V}^\top \\ \tilde{V}_r^\top \end{pmatrix} V_{1\perp}\right\|_F$$

$$= \left\|U_{1\perp}^\top \tilde{U} D_{\hat{r}} \tilde{V}^\top V_{1\perp} + U_{1\perp}^\top \tilde{U}_r D_{r-\hat{r}} \tilde{V}_r^\top V_{1\perp}\right\|_F$$

$$\leq \left\|U_{1\perp}^\top \tilde{U}\right\|_F \left\|\tilde{V}^\top V_{1\perp}\right\|_F \cdot \|D_{\hat{r}}\|_{op} + \left\|U_{1\perp}^\top \tilde{U}_r\right\|_F \left\|\tilde{V}_r^\top V_{1\perp}\right\|_F \cdot \|D_{r-\hat{r}}\|_{op}$$

$$\leq \|\Theta^*\|_{op} \cdot \frac{4\left\|\widehat{\Theta} - \Theta^*\right\|_F^2}{D_{rr}^2} + \sqrt{r - \hat{r}}^2 \cdot 2R(r+1)$$

$$\tilde{O}\left(\frac{rd^2}{D_{rr}^2}\left(\frac{d}{H}\right)^{\frac{2\delta}{1+\delta}} + r^{\frac{3}{2}}d\left(\frac{d}{H}\right)^{\frac{\delta}{1+\delta}}\right) \asymp \tilde{O}\left(r^{\frac{3}{2}}d\left(\frac{d}{H}\right)^{\frac{\delta}{1+\delta}}\right).$$

Note the second term will be dominant for large $H$, s.t. $H \geq \frac{d^{\frac{1+2\delta}{\delta}}}{r^{\frac{1+\delta}{2\delta}} D_{rr}^{\frac{2+2\delta}{\delta}}}$.

By using $T_1 = \min\left\{d \cdot 2^{\frac{i(1+\delta)}{1+2\delta}}, 2^i\right\}$ at each batch in line 3 of Algorithm 1, we can identically prove Theorem 4.4 with the same procedure as the proof of Theorem 4.3. And the only slight difference lies in the control of the term $A_{i,5}$. Therefore, we will omit the redundant details here.

$\square$

# F PROOF OF THEOREM 5.1

In this section, we will present a regret lower bound for the LowHTR. Our proof relies on the following Lemma for the MAB with heavy-tailed rewards:

**Lemma F.1.** [Xue et al., 2020] *For any multi-armed bandit algorithm $\mathcal{B}$ with $T \geq K \geq 4$ where $K$ is the number of arms, an arm $a^* \in \{1, \ldots, K\}$ is chosen uniformly at random, this arm pays $1/\gamma$ with probability $p(a^*) = 2\gamma^{1+\delta}$ and the rest pays $1/\gamma$ with probability $\gamma^{1+\delta}$ $(2\gamma^{1+\delta} < 1)$. If we set $\gamma = (K/(T+2K))^{\frac{1}{1+\delta}}$, and denote $r_{t,a}$ as the observed reward of arm $a$ at round $t$ under algorithm $\mathcal{B}$, we have*

$$\mathbb{E}\left[\sum_{t=1}^{T} r_{t,a^*} - \sum_{t=1}^{T} r_{t,a_t}\right] \geq \frac{1}{8} T^{\frac{1}{1+\delta}} K^{\frac{\delta}{1+\delta}}.$$

Therefore, we can naturally consider the LowHTR problem with a finite and fixed arm set of size $K$. For simplicity, we set $d_1 = d_2 = d$ and set $K = (d-1)r \geq 4$. To adapt the results from Lemma F.1, we make the reward function of an arm $X_{t,a} \in \mathbb{R}^{d^2}$ as

$$r_{t,a} = \begin{cases} \frac{1}{\gamma}, & \text{with probability } \gamma \cdot \langle X_{t,a}, \Theta^* \rangle \\ 0, & \text{with probability } 1 - \gamma \cdot \langle X_{t,a}, \Theta^* \rangle \end{cases},$$

and then we only need to make $\langle X_{t,a^*}, \Theta^* \rangle = 2\gamma^\delta$ and $\langle X_{t,a}, \Theta^* \rangle = \gamma^\delta$ for any other arm $a$ where $a^*$ is uniformly chosen from $[K]$.

The contextual matrices are designed in the following way. For the first column, the first $r$ entries are set to be $\left[\sqrt{\frac{1}{r(r+1)}}, \sqrt{\frac{2}{r(r+1)}}, \cdots, \sqrt{\frac{r}{r(r+1)}}\right]$. And for the rest $(d-1)r$ entries in the first $r$ rows, we flatten them and set the $i$-th entry as $\frac{1}{\sqrt{2}}$ for the $i$-th arm matrix. All the other elements in the last $(d-k)$ rows are set to null for all arm matrices. We can easily check that the Frobenious norm of all arm matrices are bounded by 1.

Next, we consider the parameter matrix $\Theta^*$ of rank $r$. For the first column, the first $r$ entries are set to be $\left[\sqrt{\frac{4}{r(r+1)}}\gamma^\delta, \sqrt{\frac{8}{r(r+1)}}\gamma^\delta, \cdots, \sqrt{\frac{4r}{r(r+1)}}\gamma^\delta\right]$. And similarly for the rest $(d-1)r$ entries in the first $r$ rows, we flatten them and uniformly choose an index from $[(d-1)r]$, then the corresponding entry is $\sqrt{2}\gamma^\delta$ and all the rest elements in $\Theta^*$ are 0. The norm of $\Theta^*$ can also be bounded with large $T$. By using the feature matrices and the parameter matrix described above, we can recover the scenario in Lemma F.1, and thus we have that

$$\mathbb{E}R(T) \geq \frac{1}{8} T^{\frac{1}{1+\delta}}(d-1)^{\frac{\delta}{1+\delta}} r^{\frac{\delta}{1+\delta}} \asymp T^{\frac{1}{1+\delta}} d^{\frac{\delta}{1+\delta}} r^{\frac{\delta}{1+\delta}} \gtrsim T^{\frac{1}{1+\delta}}.$$

$\square$

# G REMARKS OF ASSUMPTION 3.2

We will show that when a series of iid random matrices $X_{i\,i=1}^{m}$ follows a sub-Gaussian distribution with parameter $\sigma \asymp \frac{1}{\sqrt{d_1 d_2}}$, then the scale of $\max_{i \in [m]} \|X_i\|_F$ can be bounded by some constant up to some very small logarithmic terms. The results can be directly deduced from the following Lemma:

**Lemma G.1.** *If iid random matrices $X_{i\,i=1}^{m} \in \mathbb{R}^{d_1 \times d_2}$ follows a sub-Gaussian distribution with parameter $\sigma$, then with probability at least $1 - \delta$ it holds that:*

$$\|X_i\|_F \leq 4\sigma\sqrt{d_1 d_2} + 2\sqrt{2}\sigma\sqrt{\ln\left(\frac{m}{\delta}\right)}, \quad \forall i \in [m].$$

*Proof.* Denote $\mathcal{N}_{\frac{1}{2}}$ as the $\frac{1}{2}$-covering of the matrix space $\{X : \|X\|_F \leq 1\}$, then it holds that $|\mathcal{N}_{\frac{1}{2}}| \leq (1 + 1/0.5)^{d_1 d_2} =$

$5^{d_1 d_2}$. And for $\|V\|_{\mathrm{F}} \leq 1$ we define $S(V)$ as the closest point in $\mathcal{N}_{\frac{1}{2}}$ such that $\|V - S(V)\|_{\mathrm{F}} \leq \frac{1}{2}$. Next, we can have that

$$\|X_i\|_{\mathrm{F}} = \max_{\|V\|_{\mathrm{F}}=1} \langle V, X_i \rangle = \max_{\|V\|_{\mathrm{F}}=1} \langle V - S(V) + S(V), X_i \rangle \leq \max_{Z \in \mathcal{N}_{\frac{1}{2}}} \langle Z, X_i \rangle + \max_{\|W\|_{\mathrm{F}}=\frac{1}{2}} \langle W, X_i \rangle$$

$$\leq \max_{Z \in \mathcal{N}_{\frac{1}{2}}} \langle Z, X_i \rangle + \frac{1}{2} \max_{\|W\|_{\mathrm{F}}=1} \langle W, X_i \rangle,$$

which indicates that $\|X_i\|_{\mathrm{F}} \leq 2 \max_{Z \in \mathcal{N}_{\frac{1}{2}}} \langle Z, X_i \rangle$. Therefore, it holds that for any $t > 0$

$$\mathbb{P}\left(\|X_i\|_{\mathrm{F}} \geq t\right) \leq \mathbb{P}\left(\max_{Z \in \mathcal{N}_{\frac{1}{2}}} \langle Z, X_i \rangle \geq \frac{1}{2}\right) \leq |\mathcal{N}_{\frac{1}{2}}| \cdot \exp\left(-\frac{t^2}{8\sigma^2}\right) \leq 5^{d_1 d_2} \cdot \exp\left(-\frac{t^2}{8\sigma^2}\right).$$

This fact indicates that

$$\mathbb{P}\left(\|X_i\|_{\mathrm{F}} \geq 2\sqrt{2}\sigma\sqrt{\ln\left(\frac{1}{\delta}\right)} + 4\sigma\sqrt{d_1 d_2}\right) \leq \delta.$$

Therefore, we have that

$$\mathbb{P}\left(\max_{i \in [m]} \|X_i\|_{\mathrm{F}} < 2\sqrt{2}\sigma\sqrt{\ln\left(\frac{1}{\alpha}\right)} + 4\sigma\sqrt{d_1 d_2}\right) \geq (1-\alpha)^m = 1 - \delta, \quad \text{where } \alpha = 1 - (1-\delta)^{\frac{1}{m}}.$$

For any $m > 1$ and $x \in [0,1]$, based on the taylor series of the function $f(x) = (1-x)^{\frac{1}{m}} = 1 - \frac{x}{m} - O(x^2)$, it holds that $1 - \frac{x}{m} > (1-x)^{\frac{1}{m}}$. And this fact leads to the final result:

$$\mathbb{P}\left(\max_{i \in [T]} \|X_i\|_{\mathrm{F}} < 2\sqrt{2}\sigma\sqrt{\ln\left(\frac{T}{\delta}\right)} + 4\sigma\sqrt{d_1 d_2}\right) > 1 - \delta,$$

which indicates that $\max_{i \in [T]} \|X_i\|_{\mathrm{F}}$ can be uniformly bounded by a constant scale up to some minimal error. $\qquad\square$

In our case with $\sigma \asymp \frac{1}{\sqrt{d_1 d_2}}$, with probability at least $1 - \delta$ it holds that

$$\max_{i \in [m]} \|X_i\|_{\mathrm{F}} \lesssim \frac{2\sqrt{2}}{\sqrt{d_1 d_2}}\sqrt{\ln\left(\frac{m}{\delta}\right)} + 4.$$

## H  ALTERNATIVE VERSION OF LOTUS

As we mention in Subsection 4.2, we also have an alternative version of our LOTUS algorithm in a more randomized manner. Specifically, at each batch, our original version illustrated in Algorithm 1 uses the static explore-then-exploit framework, where it first randomly samples some arms from the distribution $\mathcal{D}_t$ in Assumption 3.1 and then exploits the recovered low-rank subspaces with our LowTO method. However, we can mix these two exploration and exploitation steps in each batch. Specifically, we can explore by the sampling distribution $D_t$ with the probability of $T_1^i/2^i$ at each time $t$, otherwise we will conduct the subspace transformation and LowTO algorithm based on the current $\mathcal{H}_t$. The full pseudocode is presented in Algorithm 3. We can expect the same order of regret as in Theorem 4.3 based on the fact that if we do a series of iid Bernoulli trials with probability $p$ for $n$ times, then with a high probability the sum of success will be close to $np$ for large $n$ up to some logarithmic terms.

## I  DETAILS OF THE LAMM ALGORITHM

We implement the LAMM algorithm that was first proposed in [Fan et al., 2018] and recently extended to the matrix estimation setting Yu et al. [2023] for the Huber-type estimator formulated in Eqn. (2). Here we use the unified framework proposed in [Yu et al., 2023], and for the sake of completeness we will still present its details as follows:

LAMM is presented in Algorithm 4. The LAMM method is a very efficient and scalable algorithm under high-dimensional datasets, and its first crux is establishing an isotropic quadratic function that locally upper bounds the objective function

---

**Algorithm 3** Randomized LOTUS

---

**Input:** Arm set $\mathcal{X}_t$, sampling distribution $\mathcal{D}_t$, $\delta, T_0, \eta, \lambda, \{\lambda_{i,\perp}\}_{i=1}^{+\infty}$.
**Initialization:** The history buffer index set $\mathcal{H}_1 = \{\}$, the exploration buffer index set $\mathcal{H}_2 = \{\}$.

1: Pull arm $X_t \in \mathcal{X}_t$ according to $\mathcal{D}_t$ and observe payoff $y_t$. Then add $(X_t, y_t)$ into $\mathcal{H}_1$ and $\mathcal{H}_2$ for $t \leq T_0$.
2: **for** $i = 1, 2, \ldots$ until the end of iterations **do**
3:     Set the expected exploration length $T_1 = \min\left\{\left[\frac{d^{2+4\delta} r^{1+\delta}}{D_{rr}^{2+2\delta}} 2^{i(1+\delta)}\right]^{\frac{1}{1+3\delta}}, 2^i\right\}$.
4:     **for** $t = |\mathcal{H}_1| + 1 + |\mathcal{H}_1| + 2^i$ **do**
5:         **if** Randomly sample from Bernoulli$(T_1/2^i)$ and get 1 **then**
6:             Pull arm $X_t \in \mathcal{X}_t$ according to $\mathcal{D}_t$ and observe payoff $y_t$. Then add $(X_t, y_t)$ into $\mathcal{H}_1$ and $\mathcal{H}_2$
7:         **else**
8:             Obtain the estimate $\widehat{\Theta}$ based on Eqn. (3) with $\mathcal{H}_2$, where we set $\tau_i \asymp$ $\left(|\mathcal{H}_2|/(d + \ln(2^{i+1}/\epsilon))\right)^{\frac{1}{1+\delta}} c^{\frac{1}{1+\delta}}$, $\lambda_i \asymp \sigma\left((d + \ln(2^{i+1}/\epsilon))/|\mathcal{H}_2|\right)^{\frac{\delta}{1+\delta}} c^{\frac{1}{1+\delta}}$.
9:             Calculate the full SVD of $\widehat{\Theta} = [\widehat{U}, \widehat{U}_\perp]\,\widehat{D}\,[\widehat{V}, \widehat{V}_\perp]^\top$ where $\widehat{U} \in \mathbb{R}^{d_1 \times r}, \widehat{V} \in \mathbb{R}^{d_2 \times r}$.
10:             For the next round, invoke LowTO with $\delta, [\widehat{U}, \widehat{U}_\perp], [\widehat{V}, \widehat{V}_\perp], \lambda, \lambda_{i,\perp}, \mathcal{H}_1$ and obtain the updated $\mathcal{H}_1$.
11:         **end if**
12:     **end for**
13: **end for**

---

**Algorithm 4** LAMM Algorithm for the Solution to Eqn.(2)

---

**Input:** Initial $\widehat{\Theta}_0$, stopping threshold $\epsilon, \alpha_0, \psi, \lambda$.

1: **for** $i = 1, 2, \ldots$ until $\left\|\widehat{\Theta}_i - \widehat{\Theta}_{i-1}\right\|_F \leq \epsilon$ **do**
2:     Initialize $\widehat{\Theta}_i = \widehat{\Theta}_{i-1}, \alpha_i = \max(\alpha_0, \alpha_{i-1}/\psi)$ and $s_i = 0$.
3:     **while** $F(\widehat{\Theta}_i; \widehat{\Theta}_{i-1}, \alpha_i) < \hat{L}_\tau(\widehat{\Theta}_i)$ or $s_i = 0$ **do**
4:         $\widehat{\Theta}_i = S(\widehat{\Theta}_{i-1} - \alpha_i^{-1} \nabla \hat{L}_\tau(\widehat{\Theta}_{i-1}), \alpha_i^{-1}\lambda)$.
5:         $s_i = s_i + 1, \alpha_i = \psi \cdot \alpha_i$.
6:     **end while**
7: **end for**

---

$\hat{L}_\tau(\Theta)$ at each iteration until convergence. Based on the second-order Taylor expansion, given the previous estimate $\widehat{\Theta}_{t-1}$ at iteration $t-1$, we can define the quadratic function at iteration $t$ as:

$$F(\Theta; \widehat{\Theta}_{t-1}, \alpha_k) = \hat{L}_\tau(\widehat{\Theta}_{t-1}) + \langle \nabla \hat{L}_\tau(\widehat{\Theta}_{t-1}), \Theta - \widehat{\Theta}_{t-1}\rangle + \frac{\alpha_t}{2}\left\|\Theta - \widehat{\Theta}_{t-1}\right\|_F^2,$$

with some quadratic parameter $\alpha_t > 0$. This parameter needs to be sufficiently large as we illustrated above such that $\hat{L}_\tau(\widehat{\Theta}_t) \leq F(\widehat{\Theta}_t; \widehat{\Theta}_{t-1}, \alpha_t)$ holds where

$$\widehat{\Theta}_t = \arg\min_{\Theta \in R^{d_1 \times d_2}} F(\Theta; \widehat{\Theta}_{t-1}, \alpha_t) + \lambda \|\Theta\|_{\text{nuc}}.$$

We will use an iterative increment approach on $\alpha_t$ with some multiplier $\psi > 1$ to guarantee the quadratic function $F$ majorizes the objective function $\hat{L}$ at each descent. This fact ensures the descent of the objective function at each iteration with a closed-formed solution. Specifically, to minimize the penalized isotropic quadratic function, we can deduce the solution in the following ways: for $k > 0$, define the soft-thresholding operator on a diagonal matrix $\Sigma = \text{diag}(\{\sigma_i\})$ as $S(\Sigma, k) = \text{diag}(\{\max(\sigma_i - k, 0)\})$. For any general matrix $\Theta$ with its SVD decomposition as $\Theta = U\Sigma V^\top$, we write $S(\Theta, k) = US(\Sigma, k)V^\top$. Then the solution of $\widehat{\Theta}_t$ can be represented as:

$$\widehat{\Theta}_t = S(\widehat{\Theta}_{t-1} - \alpha_t^{-1}\nabla\hat{L}_\tau(\widehat{\Theta}_{t-1}), \alpha_t^{-1}\lambda).$$