# OpenReview forum: "Low-rank Matrix Bandits with Heavy-tailed Rewards"
_auai.org/UAI/2024/Conference — UAI 2024 poster_

### Official Review · Reviewer_EXWZ · 2024-03-21

**Q2-1 Originality-Novelty:** 3
**Q2-2 Correctness-Technical Quality:** 3
**Q2-5 Clarity Of Writing:** 3

**Q1 Summary And Contributions:**

This paper tackles the LowHTR (low-rank matrix bandit with heavy-tailed rewards) problem, which arises in stochastic low-rank matrix bandits with rewards following heavy-tailed distributions instead of sub-Gaussian noise. The authors propose a novel algorithm, LOTUS, to address this problem effectively. By leveraging truncation on observed payoffs and dynamic exploration, LOTUS achieves a satisfactory regret bound without requiring prior knowledge of the time horizon T. The established lower bound indicates that LOTUS is nearly optimal in terms of the order of T. Furthermore, the authors enhance LOTUS by developing a rank-free variant that eliminates the need for knowing the rank r. Simulation experiments are conducted to demonstrate the practical superiority of LOTUS, showcasing its effectiveness in solving the LowHTR problem.

**Q2-3 Extent To Which Claims Are Supported By Evidence:**

3: Good: the main claims are supported by convincing evidence (in the form of adequate experimental evaluation, proofs, (pseudo-)code, references, assumptions).

**Q2-4 Reproducibility:**

3: Good: key resources (e.g. proofs, code, data) are available and key details (e.g. proofs, experimental setup) are sufficiently well-described for competent researchers to confidently reproduce the main results.

**Q3 Main Strengths:**

1. The paper introduces the LowHTR problem, which extends the existing framework of stochastic low-rank matrix bandits by considering heavy-tailed rewards instead of the traditional assumption of sub-Gaussian noise.

2. The authors introduce the LOTUS algorithm, tailored to address the LowHTR issue. By using truncation on observed payoffs and dynamic exploration, LOTUS attains a promising regret bound, showcasing its efficacy in managing heavy-tailed rewards.

3. LOTUS achieves a regret bound that rivals the best performance seen with sub-Gaussian noise assumptions. The lower bound solidifies LOTUS's near-optimal status concerning the time horizon T.

4. The authors enhance LOTUS by developing a rank-free variant, eliminating the need for prior knowledge of the rank r.

**Q4 Main Weakness:**

1. It is unclear how to choose the parameters, such as $\tau_i, \lambda_i, b_t$

2. The rates may not be optimal in terms of $d$ and $r$

**Q5 Detailed Comments To The Authors:**

1. Figure 1 exhibits noticeable jumps in the curve for LOTUS, whereas the curve for LowESTR appears considerably smoother. What are the reasons？

2. More details should be given for the selection of parameters.

3. Just be curious, can LOTUS work for Cauchy noises?

**Q9 Complying With Reviewing Instructions:**

Yes

---

> ### Author Rebuttal · Authors · 2024-04-03
>
> Thank you very much for your insightful comments. We are happy to learn that you find our work introduces a new problem and has sound theoretical analysis. Please see our responses to your concerns:
>
> Q4.1: Thank you very much for your review. In our simulations, we set all the parameters to their theoretical values deduced in our theorems. Specifically, we set the false probability $\epsilon = 0.01$. According to Theorem 4.1, we make
> $$\tau_i = (|\mathcal{H}_2| / (d + \ln(1/\epsilon)))^\frac{1}{1+\delta} c^\frac{1}{1+\delta}, \lambda_i = \sigma ((d+\ln(1/\epsilon)) / |\mathcal{H}_2|)^\frac{\delta}{1+\delta} c^\frac{1}{1+\delta}.$$
> Where $\mathcal{H}_2$ is the exploration buffer set at each time. And based on Theorem 4.2 we set
> $$b_t = (b/\ln(2p/\epsilon))^\frac{1}{1+\delta} (t+H)^\frac{1-\delta}{2+2\delta}.$$
> Note we did not explicitly detail the parameter selection in our paper since our parameter selection is strictly aligned with the theoretical deduction, and it achieves good performance in practice as shown in Figure 1. We will elucidate this point in our revision. Thank you for your insights, which are instrumental in enhancing our work.
>
> Q4.2: We acknowledge that the rate may not be optimal in terms of $d$ and $r$. On the one hand, we believe the time horizon $T$ is the most significant value in magnitude, and we have achieved the optimal regret bound in terms of $T$. On the other hand, note even for the simpler low-rank matrix bandits under sub-Gaussian noise, there remains a gap for $d$ and $r$ between the lower bound deduced in Lu et al. 2021 and the state-of-the-art upper bounds deduced in Lu et al. 2021, Kang et al. 2022. Therefore, it is intrinsically difficult to tight the order of $d$ and $r$ in the low-rank matrix bandit literature, and hence it remains a challenging future work. Note by taking $\delta = 1$, our regret bound in Theorem 4.3 matches the modern one for low-rank matrix bandits under sub-Gaussian noise deduced in Lu et al. 2021, Kang et al. 2022.
>
> Q5.1: This is because our LOTUS ran in a batch manner. At each batch, our LOTUS first randomly pulls some arms for a better parameter matrix estimation and then exploits the UCB-type LowTO algorithm. Therefore, LOTUS will first incur a linear regret bound due to random exploration at the beginning of each batch. By doing that, we can make LOTUS agnostic to the time horizon $T$. For LowESTR, it strictly requires the value of $T$ as an input parameter, and then follows an explore-and-then-exploit manner. Consequently, our LOTUS is consistently better than LowESTR under all heavy-tailed settings.
>
> Q5.2: Thank you very much. As we mentioned in our response to your Q4.1, we will clarify our parameter selection in the revision.
>
> Q5.3: Theoretically, we believe our regret bound will become futile under the Cauchy noises since its first moment is infinitely large, and hence the regret bound will become in the order of $T$. Note all the heavy-tailed bandits (Bubeck et al. 2013) will fail under Cauchy noises since the concentration results can not be deduced. Empirically, we ran our experiments under the Cauchy noises, and due to the time limit we only ran the fixed arm setting (first row in Figure 1). We set $\delta = 0.1$ and the moment bound $c=5$ in our LOTUS, and the results are shown as follows:
>
>
> | LowESTR | LOTUS   |
> |---------|---------|
> | 7425.60 | 4901.98 |
>
> According to the results, we can see that LOTUS can still yield better performance than LowESTR does, but the final cumulative regret of LOTUS is larger compared with t-distribution, Pareto and Laplace noise cases. This fact is aligned with our expectations since Cauchy noises with huge heavy-tailedness will lead to a terrible theoretical bound.
>
>
> We sincerely appreciate your careful review and value your insightful feedbacks. And we are delighted to engage in any further discussion with you to address any concern.

---

### Official Review · Reviewer_PiLS · 2024-03-23

**Q2-1 Originality-Novelty:** 3
**Q2-2 Correctness-Technical Quality:** 3
**Q2-5 Clarity Of Writing:** 3

**Q1 Summary And Contributions:**

This paper makes an interesting attempt to relax the rather strict assumption of sub-gaussian noise in payoffs and studies the case when the reward has finite $(1+\delta)$ moment instead. A new algorithm is proposed to obtain certain regret bound and a lower bound on the problem is established, suggesting the proposed algorithm is nearly optimal.

**Q2-3 Extent To Which Claims Are Supported By Evidence:**

3: Good: the main claims are supported by convincing evidence (in the form of adequate experimental evaluation, proofs, (pseudo-)code, references, assumptions).

**Q2-4 Reproducibility:**

3: Good: key resources (e.g. proofs, code, data) are available and key details (e.g. proofs, experimental setup) are sufficiently well-described for competent researchers to confidently reproduce the main results.

**Q3 Main Strengths:**

The paper studies an interesting problem in a rigorous way. The presentation is clear and makes significant contribution to the field. Different situations are properly discussed.

**Q4 Main Weakness:**

It may be worth the time to consider more than two settings in the experiment section or at least explain why it is not necessary to consider more than the two settings mentioned.

**Q5 Detailed Comments To The Authors:**

It may be worth the time to consider more than two settings in the experiment section or at least explain why it is not necessary to consider more than the two settings mentioned.

**Q9 Complying With Reviewing Instructions:**

Yes

---

> ### Author Rebuttal · Authors · 2024-04-03
>
> We are genuinely grateful for your comprehensive review of our work. And we are very happy to know that your find our work studies an interesting problem in a rigorous way with strong theoretical supports. Please see our response to your concern:
>
>
> As you mention in your insightful review, our work is mostly theoretically oriented and the experimental results mainly serve to validate the high efficiency of our proposed LOTUS as a bonus point. In our simulations, we consider three different types of heavy-tail noise, and for each type of noise, we consider two different contextual matrix settings: when the arm set is fixed or when the arm set is changing over time. Since our work mainly studies the heavy-tailed noise on bandits, we implement three classic types of heavy-tailed noise to showcase the robustness of our proposed LOTUS. Based on our knowledge, our simulations are more comprehensive compared with the state-of-the-art existing literature Lu et al. 2021 and Kang et al. 2022. Note Lu et al. 2021 simply consider one problem setting with two values of matrix rank $r$, and Kang et al. 2023 consider four types of matrix settings but with fixed noise generator. And it is quite obvious to see the practical advantages of our proposed LOTUS over LowESTR under heavy-tailed noise. We will put this detailed discussion in the revision. Thank you for helping us improve the quality of our work.
>
>
> We sincerely appreciate your valuable comments. Please let us know if you have any additional questions or concerns regarding our work.

---

### Official Review · Reviewer_rNvB · 2024-03-25

**Q2-1 Originality-Novelty:** 3
**Q2-2 Correctness-Technical Quality:** 2
**Q2-5 Clarity Of Writing:** 2

**Q1 Summary And Contributions:**

The authors introduce a novel algorithm for stochastic linear bandit problems with heavy-tailed rewards. Their algorithm achieves a regret bound that aligns with the well-studied sub-Gaussian noise scenario. Their approach remains independent of the rank of the parameter matrix. Additionally, they substantiate their claims through simulated experiments.

**Q2-3 Extent To Which Claims Are Supported By Evidence:**

2: Fair: the main claims are somewhat supported by evidence (but the experimental evaluation may be weak, or does not match entirely with the claims, important baselines may be missing, proofs contain important ideas but lack rigor, algorithmic details are only discussed superficially, references are imprecise, assumptions are not sufficiently motivated or explicated, etc.).

**Q2-4 Reproducibility:**

2: Fair: key resources (e.g. proofs, code, data) are unavailable but key details (e.g. proof sketches, experimental setup) are sufficiently well-described for an expert to confidently reproduce the main results.

**Q3 Main Strengths:**

- The problem is interesting and relevant within the stochastic linear bandits' field.
- Exploration of related work is fairly extensive.
- The theoretical results are noteworthy.
- The assumptions are fairly mild.

**Q4 Main Weakness:**

- The overall readability of the paper is not good.
- The theoretical analysis is difficult to follow.
- The experiments do not involve real-world cases.

**Q5 Detailed Comments To The Authors:**

- When the regret bounds were mentioned in the abstract and introduction, different variables occurring in there were not explained. This majorly limited the readability for me.
- It would be good to have a subsection titled Contributions.
- It would be good to have experiments for some real-world cases and not just simulations.
- Can you add more (synthetic) baselines?
- What is $|| \cdot||_F$, Frobenius norm? It would be good to explain it at least once the first time this notation was introduced.
- There are several typos. The writing needs to be improved.
- Runtime and space analysis of the algorithm should be done.

**Q9 Complying With Reviewing Instructions:**

Yes

---

> ### Author Rebuttal · Authors · 2024-04-03
>
> We sincerely appreciate your detailed review on our work. And we are more than happy to know that you find our work deduces noteworthy theoretical results with extensive exploration. Please see our responses to your questions as follows:
>
>
> Q5.1: Thank you for your feedback. We will relocate footnote 1 to the first page to ensure that all notations are clearly explained before we use them.
>
> Q5.2: Thank you for this suggestion. The contributions of our work have been summarized in the last paragraph of Section 1 Introduction before “Notations”. We will make this paragraph a small subsection to emphasize.
>
> Q5.3: Thanks for your comment. **Before we reply to your concerns about our experimental results, we’d like to first highlight that our paper is mostly theoretically oriented, and simulations can serve to further validate the high efficiency of our proposed LOTUS as a bonus.** Given the absence of real data implementation in the existing matrix bandit literature, we currently lack a real dataset for our study. However, we utilized the famous Movielens 100K dataset as our potential real dataset these days. This dataset comprises ratings for 1,682 movies from 943 users. Through data preprocessing, we aim to extract feature vectors for each movie and user using matrix factorization techniques. Subsequently, we plan to represent each user-movie pair through the outer product of their respective feature vectors. The model parameter matrix, denoted as $\Theta^*$, will be determined as the average of the feature matrices from a random selection of user-movie pairs. Due to the time limit, we implement both LOTUS with $\delta = 0.5$ and LowESTR under only the same student’s t-distributed noise in Section 6. And we fix the arm set with 500 random matrices as the same setting in Section 6. After the total time horizon T = 60,000, the cumulative regret (mean of 10 repeated experiments) is displayed in the following table:
>
>
> | LowESTR | LOTUS   |
> |---------|---------|
> | 4980.05 | 2679.47 |
>
>
> We can clearly observe that our proposed LOTUS could yield better performance.
>
> Q5.4: Since our algorithm is the first one to handle the low-rank matrix bandit under heavy-tailed noise, there is no existing baseline designed for our problem setting. Therefore, we just implement the most popular LowESTR as the baseline to showcase that the existing algorithms designed for sub-Gaussian noise will become futile when the actual noise exhibits heavy-tailedness.
>
> Q5.5: Yes, it is the Frobenius norm. We will explain it in the “Notations” part in the revision.
>
> Q5.6: Thank you very much for your careful review. We will correct our typos and clarify our notations in the revision.
>
> Q5.7: We reran the first four experiments (first two columns) these days and report their running time in minutes:
>
> | Method                           | LowESTR | LOTUS  |
> |----------------------------------|---------|--------|
> | t-distribution noise             | 309.51  | 342.99 |
> | t-distribution noise, contextual | 341.46  | 377.08 |
> | Pareto noise                     | 316.33  | 349.15 |
> | Pareto noise, contextual         | 346.70  | 382.35 |
>
> From the table we can observe that LOTUS is slightly more computationally expensive than LowESTR since the heavy-tailed noise is much more difficult to handle, while the performance of LOTUS is significantly better than LowESTR as shown in our paper.
>
> Thank you very much for your valuable comments. Please let us know if our response has decently resolved your concern and improved your opinion of our work. And we are more than happy to take any additional questions from you.

---

### Official Review · Reviewer_DufR · 2024-03-25

**Q2-1 Originality-Novelty:** 2
**Q2-2 Correctness-Technical Quality:** 2
**Q2-5 Clarity Of Writing:** 2

**Q1 Summary And Contributions:**

This work studied the low-rank matrix bandits with heavy-tailed rewards. It proposed the LOTUS algorithm to solve this problem while the rank $r$ is known or unknown. It analyzed the regret of the LOTUS algorithm under these assumptions. It also evaluated the performance of LOTUS with numerical experiments.

**Q2-3 Extent To Which Claims Are Supported By Evidence:**

2: Fair: the main claims are somewhat supported by evidence (but the experimental evaluation may be weak, or does not match entirely with the claims, important baselines may be missing, proofs contain important ideas but lack rigor, algorithmic details are only discussed superficially, references are imprecise, assumptions are not sufficiently motivated or explicated, etc.).

**Q2-4 Reproducibility:**

3: Good: key resources (e.g. proofs, code, data) are available and key details (e.g. proofs, experimental setup) are sufficiently well-described for competent researchers to confidently reproduce the main results.

**Q3 Main Strengths:**

1. The low-rank bandit with heavy-tailed bounds is a practical setting. I appreciate the author(s) focus on this topic.
1. The author(s) discussed an amount of related works and also some key techniques in existing literature.
1. The author(s) considered the difficult case where the rank $r$ is unknown.

**Q4 Main Weakness:**

1. Regarding the parameter $\delta$ in the assumption of moment. Does it equal to $1$ in existing work on low-rank matrix bandits?
1. The author(s) may consider to present the existing results in bandits with heavy-tails / low-rank matrix bandits in two separate tables? That will ease the comparison of existing works and the readers can have a better understanding of existing results.
1. What is the technical results in the works on 'matrix recovery under heavy-tailedness'?
1. The discussion after Assumption 3.1 should be improved to support the fairness of Assumption 3.1:
      1. The statement '... we can find such a sampling distribution if the convex hull of this region contains a ball with some constant radius'. How? I appreciate some evidence of this statement.

      1. It takes the sub-Gaussian case as an instance to support the popularity of Assumption 3.1. However, I think the sub-Gaussian case does not fit in the focus of 'heavy-tailed' in this paper. Besides, the two reference here does not focus on 'heavy-tailed'.
1. The symbols $\gtrsim$, $ \asymp $ are used throughout the paper. I feel they are not commonly used in a paper focused on analysis and at least their meanings should be explained. Besides, the author(s) should accurately describe the relation between values. For instance:
      1. In Assumption 3.1, is $\sigma^2$ or $c_l$ larger? Can their relationship be described with an equation/inequality?

      1. In Theorem 4.1, can be relationship between $n$ and $drv^3$ be described with an inequality?

      1. After Theorem 4.3: $d\gtrsim T^{\cdot}$
1. Section 4 - 'Contracting the results in [Yu et al., 2023], we further prove that our Huber-type estimator is robust to arbitrary heavy-tailed noise with the finite $(1 + \delta)$ moment for $\delta \in (0, 1)$ on the trace regression problem.' What is the assumption in Yu et al. [2013]?
1. Section 4.2: The parameter $T_1^i$ is set with $D_{rr}$. As $\Theta^*$ is unknown, why is the matrix $D$ known by the agent?
1. The LOTUS algorithm is claimed to be possibly run in a more randomized manner. May the author(s) provide some evidence? Is there any benefit of running it more randomly?
1. The sampling distribution $\mathcal{D}_t$ is an input to the LOTUS algorithm in Line 1 of Algorithm 1. What is that? Instead, should the algorithm generate the sampling distribution adaptively?
1. The author(s) should highlight where the Algorithms 2 and 4 are utilized in Algorithm 1. For instance, Line 7 may be stated as '... invoke LowTO (presented in Algorithm 2) with $\delta$ ...'
1. Beginning of Section 4.3: 'this information is never revealed to agents in real-world applications'. This statement is a bit strong. I suggest the author(s) to provide a milder claim.
1. Is there an existing lower bound in the low-rank bandits? Is the proposed lower bound in the heavy-tailed setting reduced to an existing bound in low-rank bandits when $\delta=1$?

**Q5 Detailed Comments To The Authors:**

My key concerns are listed in Q4 and below (due to character constraint of Q4):
1. The LowGLOC algorithm seems to highly related to the setting. May the author(s) include it in the experiment setting?
1. The LOTUS algorithm can be applied in the low-rank bandits with light/heavy tails. In that case, the theoretical and numerical performance should be compared to existing algorithms in low-rank bandits with 'light tails'. Maybe when $\delta=1$?

Besides, some other minor suggestions are as below:
1. Top-right corner in page 4: '[Fan et al. [2018], Sun et al. [2020], Yu et al. [2023]' should be changed to '[Fan et al. 2018, Sun et al. 2020, Yu et al. 2023]'.
1. Section 4.2: 'As shown in Algorithm 1 line 8' should be revised to be 'As shown in Line 8 of Algorithm 1'. Some similar changes should also be made.

**Q9 Complying With Reviewing Instructions:**

Yes

---

> ### Author Rebuttal · Authors · 2024-04-03
>
> Thank you for your valuable comments. Please see our responses to your concerns. We have to omit responses to typos/writing due to space limit, and we will correct the typos and clarify our notations in the revision.
>
> 4.1 As we mention in Section 2, all low-rank matrix bandits assume the noise is sub-Gaussian, which implies the noise has a finite moment of any order. By setting $\delta = 1$, we assume the noise only has second-order moment, which is a strictly weaker condition.
>
> 4.3 We didn’t elaborate on matrix recovery under heavy-tailedness since this is not related to bandits and due to the space limit. For the two works mostly related to our trace regression problem, [3] achieves the error rate of order $\tilde O(\sqrt{\frac{rd}{n}})$ when the noise possesses finite $2k$ moment for $k > 1$. [4] improves their result and their algorithm can achieve the error rate of order $\tilde O(\sqrt{\frac{rdv}{n}})$ when the noise has finite variance bounded by $v$. But neither of them considers the more challenging setting of arbitrary heavy-tailed noise with only bounded $(1+\delta)$ moment, $\delta \in (0,1)$. Note our convergence rate in Theorem 4.1 matches the rate in [4] with $\delta = 1$.
>
> 4.4.1 We can simply use the uniform distribution on the ball with a constant radius, and this sampling will be sub-Gaussian with parameter in the scale of $1/(d_1d_2)$. Note this is a default statement in low-rank matrix bandit and high-dimensional statistics, and both [1] (after Assumption 2) and [2] (after Assumption 3.5) mentioned it in their work. We will put a detailed explanation on this statement in the revision. 4.4.2 We assume the heavy-tailedness on the noises $\eta_t$ but not the contextual matrices as we present in Section 3. Assumption 3.1 is about sampling distribution on the contextual matrices but not the rewards, and this assumption is standard in the existing literature ([1], [2], [3]).
>
> 4.5 Symbol $\asymp$ denotes “two values are asymptotically equivalent”, i.e. $a \asymp b$ is identical to $a = \Theta(b)$. While $\gtrsim$ means “greater than or approximately equal to”, i.e. $a \gtrsim b$ is identical to $b = O(a)$, these two notations are widely used in high-dimensional analysis ([1]-[4]) and we will clarify them in our revision. (2) the exact relationship is deduced in Theorem A.6 in Appendix. 5.3.
>
> 4.6 As we mention in Section 2, [4] assume the noise has a bounded variance, i.e. $\delta = 1$.
>
> 4.7 It is essential to note that there are problem-dependent factors unknown to the algorithm for all low-rank matrix bandits or even any bandit algorithms. For low-rank matrix bandits, existing approaches ([1], [2]) encounter the same problem and their implementation requires the value of $D_{rr}$ as well. Our method doesn't rely on any additional parameters to tackle the more challenging LowHTR problem. For other bandit algorithms, most works assume the noise is sub-Gaussian($s^2$) for some unknown $s$, but these algorithms always require the value of $s$.
>
> 4.8 The randomized LOTUS is presented in Algorithm 3 in Appendix. From our knowledge, randomized version will not bring any benefits, and hence we just present it in Appendix for completeness of our work.
>
> 4.9 Sampling distribution $D_t$ is used to draw arms at the beginning of each batch for exploration. Its property is defined in Assumption 3. Note both LowESTR and G-ESTT in [1] and [2] also require this distribution for warm-up.
>
> 4.12 There is an existing lower bound under sub-Gaussian noise in [1] in the order of $\tilde O (dr \sqrt{T})$. Our deduced regret bound matches this lower bound with $\delta=1$ in terms of $T$. Note $\delta = 1$ condition is strictly weaker than the sub-Gaussian condition, so the results under these two cases may not be directly comparable.
>
> 5.1 As mentioned by the authors of [1] in their Section 6, LowGLOC needs to calculate the weights of the covering of low-rank matrices, whereas it is unknown how to find the exact covering of low-rank matrices and the calculation of the weights is also computationally formidable. In other words, LowGLOC is proposed purely for theoretical analysis, and hence it has never been implemented in neither its original paper nor any follow-up work. Therefore, we use the efficienct LowESTR as the baseline as suggested by [1].
>
> 5.2 Theoretically, as we mention after Theorem 4.3, LOTUS could achieve the same regret bound when $\delta=1$ as the state-of-the-art regret bounds proposed in [1], [2] under sub-Gaussian noise. Empirically, we utilize the Laplace noise with finite variance ($\delta = 1$) shown in the last column of Figure 1. We can observe that LOTUS yields better performance under “light-tail” noise.
>
> We sincerely appreciate your efforts and time. The score is a bit low even though no critical issue remains (especially with rebuttal). Please let us know if our response has improved your opinion of our work.
>
> [1] Lu et al. 2021
>
> [2] Kang et al. 2022
>
> [3] Fan et al, 2021
>
> [4] Yu et al. 2023

---

### Official Review · Reviewer_TYVK · 2024-03-27

**Q2-1 Originality-Novelty:** 3
**Q2-2 Correctness-Technical Quality:** 3
**Q2-5 Clarity Of Writing:** 3

**Q10 Ethical Concerns:**

No.

**Q1 Summary And Contributions:**

The paper considers the problem of low-rank matrix bandits with heavy tailed rewards. The proposed LOTUS algorithm attains a regret bound of $\tilde{O}(T^{1/1+\delta})$, which matches the lower bound they established.

**Q2-3 Extent To Which Claims Are Supported By Evidence:**

3: Good: the main claims are supported by convincing evidence (in the form of adequate experimental evaluation, proofs, (pseudo-)code, references, assumptions).

**Q2-4 Reproducibility:**

3: Good: key resources (e.g. proofs, code, data) are available and key details (e.g. proofs, experimental setup) are sufficiently well-described for competent researchers to confidently reproduce the main results.

**Q3 Main Strengths:**

The paper is well written. The paper has made comprehensive discussions on literature reviews and thorough comparisons with existing methods. Both theoretical results and numerical results are clearly stated and justified.

**Q4 Main Weakness:**

I did not spot any major weakness of the paper.

**Q5 Detailed Comments To The Authors:**

While the subroutine is based on explore-then-commit style, is it possible to replace it with better algorithms like UCB to improve regret bounds?

**Q9 Complying With Reviewing Instructions:**

Yes

---

> ### Author Rebuttal · Authors · 2024-04-03
>
> Thank you very much for your careful review. We are more than happy to know that you find our work is well written with a comprehensive discussion on existing methods. And our work is sound in both theory and practice.
>
> Q5: For the high-dimensional bandit problem, since there are too many covariates to be estimated in the beginning, it is more efficient to first have a rough estimation of the high-dimensional structure and then use UCB-type algorithm to refine our estimate while balancing the exploration-exploitation trade-off. And this is exactly what our Algorithm 1 does as shown in its pseudocode. Specifically, our Algorithm 2 (LowTO) is a UCB-based algorithm, and we invoke LowTO after the exploration phase at each batch of LOTUS (line 7 of Algorithm 1).
>
> We sincerely appreciate your thoughtful review, and we are more than delighted to engage in further discussions with you and take any additional questions from you.

---

### Meta-Review · Area_Chair_3kgZ · 2024-04-17

Most reviewers found this paper's results interesting. The authors provided a comprehensive rebuttal to Reviewer DufR's questions.